# Inference and analysis of cell-cell communication using CellChat

Suoqin Jin [1,2], Christian F. Guerrero-Juarez [1,2,3,4], Lihua Zhang[1,2], Ivan Chang[5,6], Raul Ramos[2,3,4], Chen-Hsiang Kuan[3,4,7,8], Peggy Myung [9,10], Maksim V. Plikus [2,3,4 ✉] & Qing Nie [1,2,3 ✉]

Understanding global communications among cells requires accurate representation of cell-cell signaling links and effective systems-level analyses of those links. We construct a database of interactions among ligands, receptors and their cofactors that accurately represent known heteromeric molecular complexes. We then develop CellChat, a tool that is able to quantitatively infer and analyze intercellular communication networks from single-cell RNA-sequencing (scRNA-seq) data. CellChat predicts major signaling inputs and outputs for cells and how those cells and signals coordinate for functions using network analysis and pattern recognition approaches. Through manifold learning and quantitative contrasts, Cell-Chat classifies signaling pathways and delineates conserved and context-specific pathways across different datasets. Applying CellChat to mouse and human skin datasets shows its ability to extract complex signaling patterns. Our versatile and easy-to-use toolkit CellChat and a web-based Explorer (http://www.cellchat.org/) will help discover novel intercellular communications and build cell-cell communication atlases in diverse tissues.

[1] Department of Mathematics, University of California, Irvine, Irvine, CA, USA. [2] NSF-Simons Center for Multiscale Cell Fate Research, University of California, Irvine, Irvine, CA, USA. [3] Department of Developmental and Cell Biology, University of California, Irvine, Irvine, CA, USA. [4] Sue and Bill Gross Stem Cell Research Center, University of California, Irvine, Irvine, CA, USA. [5] Department of Biological Chemistry, University of California, Irvine, Irvine, CA, USA. [6] Research Cyberinfrastructure Center, University of California, Irvine, Irvine, CA, USA. [7] Graduate Institute of Clinical Medicine, College of Medicine, National Taiwan University, Taipei, Taiwan. [8] Division of Plastic Surgery, Department of Surgery, National Taiwan University, Taipei, Taiwan. [9] Department of Dermatology, Yale University, New Haven, CT, USA. [10] Department of Pathology, Yale University, New Haven, CT, USA. ✉email: plikus@uci.edu; qnie@uci.edu

Signaling crosstalk via soluble and membrane-bound factors is critical for informing diverse cellular decisions, including decisions to activate cell cycle or programmed cell death, undergo migration or differentiate along the lineage[1–3]. Single-cell RNA-sequencing (scRNA-seq) technologies have led to discovery of cellular heterogeneity and differentiation trajectories at unprecedented resolution level[4,5]. While most current scRNA-seq data analysis approaches allow detailed cataloging of cell types and prediction of cellular differentiation trajectories, they have limited capability in probing underlying intercellular communications that often drive heterogeneity and cell state transitions. Yet, scRNA-seq data inherently contains gene expression information that could be used to infer such intercellular communications[6,7].

Several methods have been recently developed to infer cell–cell communication from scRNA-seq data[8–14], such as Single-CellSignalR[9], iTALK[10], and NicheNet[13]. However, these and other similar methods usually use only one ligand/one receptor gene pairs, often neglecting that many receptors function as multi-subunit complexes. For example, soluble ligands from the TGFβ pathway signal via heteromeric complexes of type I and type II receptors[15]. More recently, to address this limitation, CellPhoneDB v2.0 has been developed, which predicts enriched signaling interactions between two cell populations by considering the minimum average expression of the members of the heteromeric complex[16]. However, it does so without considering other important signaling cofactors, including soluble agonists, antagonist, as well as stimulatory and inhibitory membrane-bound co-receptors. Other limitations of current databases or tools include the lack of: (a) systematically curated classification of ligand-receptor pairs into functionally related signaling pathways; (b) intuitive visualization of both autocrine and paracrine signaling interactions; (c) systems approaches for analyzing complex cell–cell communication; and (d) capability of accessing signaling crosstalk for continuous cell state trajectories given the fact that biological variability between cells can be discrete or continuous.

Here we develop CellChat, an open source R package (https://github.com/sqjin/CellChat) to infer, visualize and analyze intercellular communications from scRNA-seq data. First, we manually curate a comprehensive signaling molecule interaction database that takes into account the known structural composition of ligand-receptor interactions, such as multimeric ligand-receptor complexes, soluble agonists and antagonists, as well as stimulatory and inhibitory membrane-bound co-receptors. Next, CellChat infers cell-state specific signaling communications within a given scRNA-seq data using mass action models, along with differential expression analysis and statistical tests on cell groups, which can be both discrete states or continuous states along the pseudotime cell trajectory. CellChat also provides several visualization outputs to facilitate intuitive user-guided data interpretation. CellChat can quantitatively characterize and compare the inferred intercellular communications through social network analysis tool[17], pattern recognition methods[18,19] and manifold learning approaches[20]. Such analyses enable identification of the specific signaling roles played by each cell population, as well as generalizable rules of intercellular communications within complex tissues. We showcase CellChat's overall capabilities by applying it to both our own and publicly deposited mouse skin scRNA-seq datasets from embryonic development and adult wound healing stages, as well as human skin scRNA-seq dataset from a diseased state. A systematic comparison with several existing tools for cell–cell communication is also presented.

## Results

**Overview of CellChat.** CellChat requires gene expression data from cells as the user input and models the probability of cell–cell communication by integrating gene expression with prior knowledge of the interactions between signaling ligands, receptors and their cofactors (Fig. 1a). To establish intercellular communications, CellChat can operate in label-based and label-free modes (Fig. 1b). In its label-based mode, CellChat requires user-assigned cell labels as the input. In its label-free mode, CellChat requires user input in form of a low-dimensional representation of the data, such as principal component analysis or diffusion map. For the latter, CellChat automatically groups cells by building a shared neighbor graph based on the cell–cell distance in the low-dimensional space or the pseudotemporal trajectory space (see "Methods" section). Upon receiving input data, Cell-Chat models intercellular communications via the following three modules:

Cross-referencing ligand-receptor interaction database. The accuracy of the assigned roles for the signaling molecules and their interactions is crucial for predicting biologically meaningful intercellular communications. We manually curated a literature-supported signaling molecule interaction database, called Cell-ChatDB, which takes into account the known composition of ligand-receptor complexes, including complexes with multimeric ligands and receptors, as well as several cofactors: soluble agonists, antagonists, co-stimulatory and co-inhibitory membrane-bound receptors (Fig. 1a, Supplementary Fig. 1a, Supplementary Note 1). CellChatDB incorporates signaling molecule interaction information from the KEGG Pathway database[21], a collection of manually drawn signaling pathway maps assembled by expert curators based on existing literature. It also includes information from recent experimental studies. CellChatDB contains 2,021 validated molecular interactions, including 60% of paracrine/autocrine signaling interactions, 21% of extracellular matrix (ECM)-receptor interactions and 19% of cell–cell contact interactions. 48% of the interactions involve heteromeric molecular complexes and 25% of the interactions are curated by us from recent literature (Fig. 1a). Furthermore, each interaction is manually classified into one of the 229 functionally related signaling pathways based on the literature.

Inference and visualization of intercellular communications. To predict significant communications, CellChat identifies differentially over-expressed ligands and receptors for each cell group (Fig. 1b; also see "Methods" section). To quantify communications between two cell groups mediated by these signaling genes, CellChat associates each interaction with a probability value. The latter is modeled by the law of mass action based on the average expression values of a ligand by one cell group and that of a receptor by another cell group, as well as their cofactors (see "Methods" scetion). Significant interactions are identified on the basis of a statistical test that randomly permutes the group labels of cells and then recalculates the interaction probability (Fig. 1c, see "Methods" section). An intercellular communication network is a weighted directed graph composed of significant connections between interacting cell groups. CellChat also provides an informative and intuitive visualization method, called hierarchical plot, to highlight autocrine and paracrine signaling communications between cell groups of interest. This hierarchical plot provides an overview of inferred intercellular communication network for each signaling pathway or ligand-receptor pair, consisting of two components: the left portion shows autocrine and paracrine signaling to certain cell groups of interest, and the right portion shows autocrine and paracrine signaling to the remaining cell groups in the dataset. In addition, CellChat implements several other visualization ways, including circle plot and bubble plot (Fig. 1d, see "Methods" section).

Quantitative analysis of intercellular communications. To facilitate the interpretation of the complex intercellular

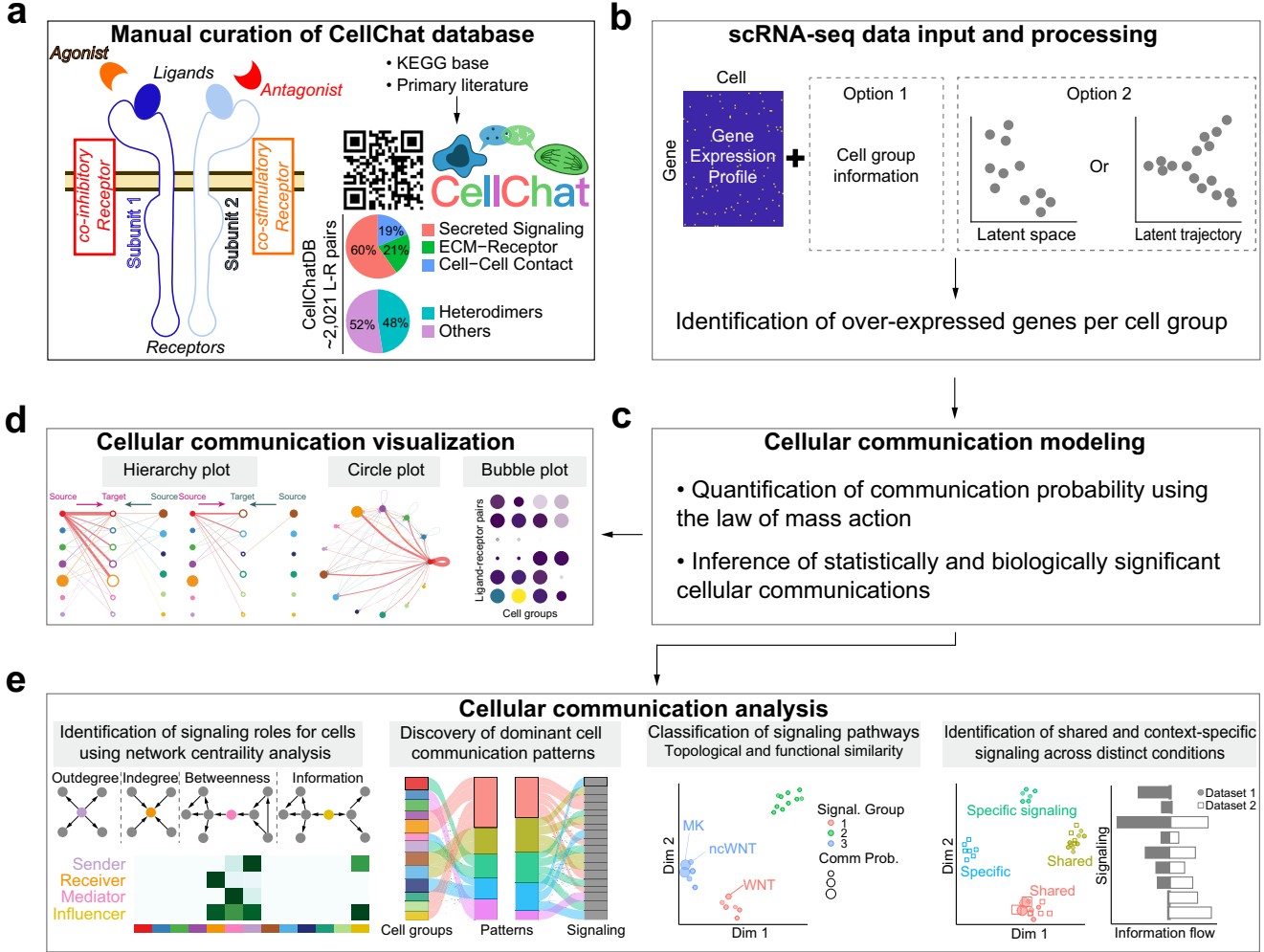

**Fig. 1 Overview of CellChat. a** Overview of the ligand-receptor interaction database. CellChatDB takes into account known composition of the ligand-receptor complexes, including complexes with multimeric ligands and receptors, as well as several cofactor types: soluble agonists, antagonists, co-stimulatory and co-inhibitory membrane-bound receptors. CellChatDB contains 2021 validated interactions, including 60% of secreting interactions. In addition, 48% of the interactions involve heteromeric molecular complexes. **b** CellChat either requires user assigned cell labels as input or automatically groups cells based on the low-dimensional data representation supplied as input. **c** CellChat models the communication probability and identifies significant communications. **d** CellChat offers several visualization outputs for different analytical tasks. Different colors in the hierarchy plot and circle plot represent different cell groups. Colors in the bubble plot are proportional to the communication probability, where dark and yellow colors correspond to the smallest and largest values. **e** CellChat quantitatively measures networks through approaches from graph theory, pattern recognition and manifold learning, to better facilitate the interpretation of intercellular communication networks and the identification of design principles. In addition to analyzing individual dataset, CellChat also delineates signaling changes across different contexts, such as different developmental stages and biological conditions.

communication networks, CellChat quantitatively measures networks through methods abstracted from graph theory, pattern recognition and manifold learning (see "Methods" section). CellChat performs a variety of analyses in an unsupervised manner (Fig. 1e). First, it can determine major signaling sources and targets, as well as mediators and influencers within a given signaling network using centrality measures from network analysis, such as out-degree, in-degree, betweenness, and information metrics (see "Methods" section). Second, it can predict key incoming and outgoing signals for specific cell types, as well as coordinated responses among different cell types by leveraging pattern recognition approaches. Outgoing patterns reveal how the sender cells (i.e., cells as signal source) coordinate with each other, as well as how they coordinate with certain signaling pathways to drive communication. Incoming patterns show how the target cells (i.e., cells as signal receivers) coordinate with each other, as well as how they coordinate with certain signaling pathways to respond to incoming signals. Third, it can

group signaling pathways by defining similarity measures and performing manifold learning from both functional and topological perspectives. Fourth, it can delineate conserved and context-specific signaling pathways by joint manifold learning of multiple networks across datasets. Overall, these functionalities allow CellChat to deconvolute complex intercellular communications in an easily interpretable way and predict biologically meaningful discoveries from scRNA-seq data.

**CellChat identifies communication patterns and predicts functions for poorly studied pathways**. We showcase CellChat functionalities by applying it to several recently published mouse skin scRNA-seq datasets from embryonic development[22] and adult wound healing stages[23]. Choice of skin was determined by our prior expertise on the aspects of skin morphogenesis and regeneration, its complex cellular make-up and the fact that the role of many signaling pathways in skin is well-established, which enables meaningful literature-based interpretation of a portion of

CellChat predictions. First, we ran CellChat analysis on scRNA-seq dataset for day 12 mouse skin wound tissue[23]. This dataset contains 21,898 cells, which cluster into 25 cell groups, including nine fibroblast (FIB), five myeloid (MYL) and six endothelial (ENDO) groups, as well as several other cell types such as T cells (TC), B cells (BC), dendritic cells (DC), and lymphatic endothelial cells (LYME) (Supplementary Fig. 2a–h; see "Methods" section).

CellChat detected 60 significant ligand-receptor pairs among the 25 cell groups, which were further categorized into 25 signaling pathways, including TGFβ, non-canonical WNT (ncWNT), TNF, SPP1, PTN, PDGF, CXCL, CCL, and MIF pathways. Network centrality analysis of the inferred TGFβ signaling network identified that several myeloid cell populations are the most prominent sources for TGFβ ligands acting onto fibroblasts (Fig. 2a, b). Of note one myeloid population MYL-A is also the dominant mediator, suggesting its role as a gatekeeper of cell–cell communication. These findings are consistent with the known critical role played by myeloid cells in initiating inflammation during skin wound healing and driving activation of skin-resident fibroblasts via TGFβ signaling[24–29]. Importantly, CellChat also predicted that certain endothelial cell populations, as well as several fibroblast populations, both known sources of TGFβ ligands, significantly contribute to myeloid-dominated TGFβ signal production in the wound. This reveals that TGFβ signaling network in skin wounds is complex and highly redundant with multiple ligand sources targeting large portion of wound fibroblasts. Interestingly, CellChat shows that the majority of TGFβ interactions among wound cells are paracrine, with only one fibroblast and one myeloid population demonstrating significant autocrine signaling (Fig. 2b). Notably, among all known ligand-receptor pairs, wound TGFβ signaling is dominated by *Tgfb1* ligand and its multimeric *Tgfbr1/Tgfbr2* receptor (Fig. 2c). In contrast with TGFβ, CellChat analysis of inferred ncWNT signaling network revealed its very distinct, non-redundant structure with only one ligand *(Wnt5a)* and only one population of fibroblasts (FIB-D) driving largely fibroblast-to-fibroblast, fibroblast-to-endothelial and to a lesser extent fibroblast-to-myeloid signaling (Fig. 2d–f). FIB-D cells highly expressed *Crabp1* and were enriched for cell cycle genes (Supplementary Fig. 2d), which likely represent an actively cycling subset of *Crabp1*-positive cells in upper wound dermis[23,30]. Network centrality analysis confirmed that FIB-D is a prominent influencer controlling the communications (Fig. 2e). Importantly, elevated expression of WNT5A in fibroblasts and its role in scarring has recently been reported[31–34].

In addition to exploring detailed communications for individual pathways, an important question is how multiple cell groups and signaling pathways coordinate to function. To address this question, CellChat employs a pattern recognition method based on non-negative matrix factorization to identify the global communication patterns, as well as the key signals in different cell groups (see "Methods" section). The output of this analysis is a set of the so-called communication patterns that connect cell groups with signaling pathways either in the context of outgoing signaling (i.e., treating cells as senders) or incoming signaling (i.e., treating cells as receivers). Application of this analysis uncovered five patterns for outgoing signaling (Fig. 2g) and five patterns for incoming signaling (Fig. 2h). This output, for example, reveals that a large portion of outgoing fibroblast signaling is characterized by pattern #4, which represents multiple pathways, including but not limited to ncWNT, SPP1, MK, and PROS (Fig. 2g). All of the outgoing myeloid cell signaling is characterized by pattern #2, representing such pathways as TGFβ, TNF, CSF, IL1, and RANKL. On the other hand, the communication patterns of target cells (Fig. 2h) shows that incoming fibroblast signaling is dominated by two patterns #1 and #3, which include signaling

pathways such as TGFβ and ncWNT, as well as PDGF, TNF, MK, and PTN among others. Majority of incoming myeloid cell signaling is characterized by the pattern #4, driven by CSF and CXCL pathways. Notably, both incoming and outgoing signaling by Schwann cells share the same pattern #1 with wound fibroblasts (Fig. 2g–h). These results show that: (1) two distinct cell types in the same tissue can rely on largely overlapping signaling networks; and that (2) certain cell types, such as fibroblasts, simultaneously activate multiple signaling patterns and pathways, while other cell types, such as myeloid cells or B cells, rely on fewer and more homogeneous communication patterns. Moreover, cross-referencing outgoing and incoming signaling patterns also provides a quick insight into the autocrine-acting vs. paracrine-acting pathways for a given cell type. For example, major autocrine-acting pathways between wound fibroblasts are MK, SEMA3, PROS, and ncWNT, and major paracrine-acting myeloid-to-fibroblasts pathways are TGFβ and TNF (Fig. 2g–h).

Further, CellChat is able to quantify the similarity between all significant signaling pathways and then group them based on their cellular communication network similarity. Grouping can be done either based on functional or structural similarity (see "Methods" section). Application of functional similarity grouping identified four groups of pathways (Fig. 2i). Group #1 is dominated by inflammatory pathways (e.g., TGFβ, TNF, IL, CCL) and largely represents paracrine signaling from myeloid and endothelial cells to fibroblasts. Group #2, which includes ncWNT, EGF, GAS, and PROS pathways, largely represents autocrine signaling between wound fibroblasts. Group #3, which includes CXCL and APELIN pathways, represents signaling from endothelial cells, while group #4, which includes MK, PTN, and SPP1 pathways, represents promiscuous signaling (i.e., signaling with high connectivity) and is dominated by signals from certain fibroblast populations and myeloid cells. By identifying poorly studied pathways that group together with other pathways, whose role is well known, this CellChat analysis can predict putative functions of the former. Different from grouping on the basis of functional similarity, which heavily weighs in similarity between sender and receiver cell groups, grouping based on structural similarity is primarily driven by the similarity of signaling network topology (Fig. 2j; see "Methods" section). Structural similarity grouping also identified four groups of signaling pathways (Fig. 2k). Group #1 represents pathways that have very few senders and numerous receivers, such as ncWNT; group #2 represents pathways with numerous senders and receivers, such as TGFβ and PTN; group #3 represents pathways with numerous senders and few receivers, such as CCL and IL1; and group #4 represents pathways with few senders and few receivers, such as PROS, IL2, and CXCL. Thus, grouping based on structural similarity reveals general mode of how sender and receiver cells utilize a given signaling pathway. Collectively, CellChat can identify key features of intercellular communications within a given scRNA-seq dataset and predict putative functions for poorly understood signaling pathways.

**CellChat reveals continuous cell lineage-associated signaling events.** In addition to discrete cell states, our framework can be applied to continuous cell states along the pseudotemporal trajectory (see "Methods" section). We demonstrate this utility using scRNA-seq data on embryonic day E14.5 mouse skin[22], when both dermal and epidermal cell lineages undergo rapid specification and give rise to new cell types within the developing hair follicles[22,35,36]. First, we inferred pseudotemporal trajectories for dermal and epidermal embryonic skin cells using the diffusion map approach (Fig. 3a-b; Supplementary Fig. 3a–d; see

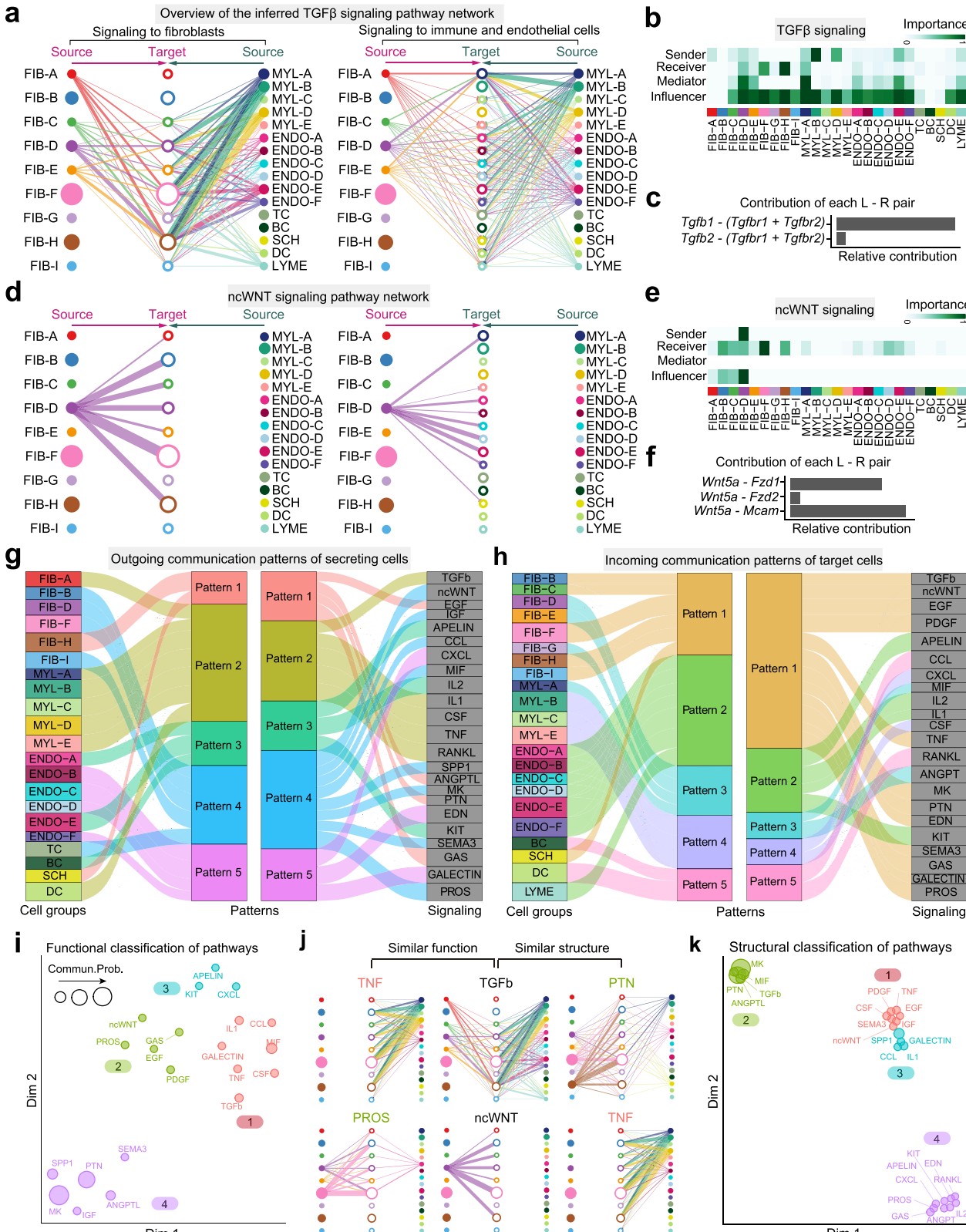

"Methods" section). Dermal cell trajectory, which on one end contains *Sox2*-high hair follicle dermal condensate (DC) cells, was divided into seven groups, that include five fibroblast states (FIB-A, FIB-B, FIB-C, FIB-D, FIB-E) and two DC states (DC-A and DC-B). A linear sequence of these trajectory groups recapitulates sequential stages of embryonic skin fibroblast lineage specification process (Fig. 3a). Embryonic epidermal cell trajectory starts from

basal epidermal cells and progresses either toward *Edar*-high epithelial placode cells or toward *Krt1*-high and *Lor*-high suprabasal epidermal cells. Collectively, epidermal lineage specification events are represented by two basal, one placode and three suprabasal trajectory groups (Fig. 3b).

We applied CellChat to study dermal-epidermal communication along these sequential cell lineage states. 88 significant

**Fig. 2 CellChat analysis of the communications between skin cells during wound repair. a** Hierarchical plot shows the inferred intercellular communication network for TGFβ signaling. This plot consists of two parts: Left and right portions highlight the autocrine and paracrine signaling to fibroblast states and to other non-fibroblast skin cell states, respectively. Solid and open circles represent source and target, respectively. Circle sizes are proportional to the number of cells in each cell group and edge width represents the communication probability. Edge colors are consistent with the signaling source. FIB-A – I: nine fibroblast cell groups; MYL-A – E: five myeloid cell groups; ENDO-A – F: six endothelial cell groups; TC: T cell; BC: B cell; SCH: Schwan cell; DC: Dendritic cell, LYME: Lymphatic endothelial cell; **(b)** Heatmap shows the relative importance of each cell group based on the computed four network centrality measures of TGFβ signaling network. **c** Relative contribution of each ligand-receptor pair to the overall communication network of TGFβ signaling pathway, which is the ratio of the total communication probability of the inferred network of each ligand-receptor pair to that of TGFβ signaling pathway. **d** The inferred ncWNT signaling network. **e** Relative contribution of each ncWNT ligand-receptor pair. **f** The computed network centrality measures of ncWNT signaling. **g** The inferred outgoing communication patterns of secreting cells, which shows the correspondence between the inferred latent patterns and cell groups, as well as signaling pathways. The thickness of the flow indicates the contribution of the cell group or signaling pathway to each latent pattern. **h** The inferred incoming communication patterns of target cells. **i** Projecting signaling pathways onto a two-dimensional manifold according to their functional similarity. Each dot represents the communication network of one signaling pathway. Dot size is proportional to the overall communication probability. Different colors represent different groups of signaling pathways. **j** Two different similarity measures are used to quantify the similarity among the inferred networks. Examples showing the functional similarity with similar major sources/targets, and structural similarity with similar network topology. **k** Projecting signaling pathways onto a two-dimensional manifold according to their structural similarity.

ligand-receptor interactions within 22 signaling pathways were predicted, including WNT, ncWNT, TGFβ, PDGF, NGF, FGF, and SEMA3. Previous studies showed that activation of canonical WNT signaling is required for DC cell specification in the embryonic skin[22,36–40]. Indeed, CellChat-inferred canonical WNT signaling network indicates that epidermal cells are the primary ligand source, which acts both in autocrine manner between epidermal cell populations, as well as in paracrine way from epidermal to dermal cells (Fig. 3c). Notably, two WNT ligand-receptor pairs, namely *Wnt6–Fzd10/Lrp6* and *Wnt6–Fzd2/Lrp6* were the dominant contributors to this communication network (Fig. 3d and Supplementary Fig. 4a), which is consistent with the previous report that *Wnt6* is the highest expressed canonical WNT ligand in embryonic mouse skin[41]. Signaling communication network for ncWNT pathway differs substantially from that of canonical WNT pathway. Late stage fibroblast state FIB-E was the primary ncWNT source, signaling both in autocrine and paracrine manner (Fig. 3e) with *Wnt5a–Fzd2* and *Wnt5–Fzd10* ligand-receptor pairs driving the signaling (Fig. 3f and Supplementary Fig. 4b–c). These results suggest distinct roles for canonical WNT and ncWNT pathways in skin morphogenesis. In another example, we analyzed the FGF signaling network (Supplementary Fig. 4d–h) and found it to be similar to the ncWNT signaling network, with the additional epithelial placode-derived *Fgf20* signaling (Supplementary Fig. 4e and h). This is consistent with the known role of placode-derived FGF20 signaling in hair follicle morphogenesis[22,42,43]. In another distinct example of TGFβ pathway, epithelial placode cells and to a lesser extent early DC-A cells were the driving sources of TGFβ ligands to dermal cells (Supplementary Fig. 4i–k). These findings are consistent with the known role for TGFβ signaling in early hair follicle morphogenesis[44,45].

We then ran CellChat pattern recognition module to uncover the key sequential signaling events along the process of skin morphogenesis. To predict the sequential signaling events, we combined the communication pattern analysis with the inferred pseudotemporal cell events. The dermal and epidermal trajectory analysis potentially revealed the pseudotemporal order of different cell types, and the communication pattern analysis identified strong signals that were sent or received by certain cell types. At the outgoing end of signaling, we predicted that FGF and GALECTIN signals are first secreted by FIB-A cells (Fig. 3g). FIB-B and FIB-C cells then coordinate production of GAS signaling. Next, FIB-D, and FIB-E fibroblasts along with suprabasal epidermal cells coordinate secretion of numerous ligands for pathways such as ncWNT, EGF, IGF, CXCL, and SEMA3; while DC-A and epithelial placode cells jointly secrete

ligands for TGFβ pathway. At the same time, basal epidermal cells dominantly drive WNT, PDGF, NGF, and VISFATIN signaling pathways. On the other hand, at the incoming end of signaling, fibroblasts are driven by patterns #1 and #2 involving pathways such as FGF, PDGF, SEMA3, TGFβ, IGF, and GALECTIN (Fig. 3h). DC and epithelial placode cells are driven by the pattern #4, which includes HH and CXCL signaling; basal epidermal cells are dominated by pattern #3 pathways—WNT, ncWNT, and EGF; while suprabasal epidermal cells are the primary target for GRN (granulin) signaling within pattern #5. Together, CellChat analysis faithfully recovers many signaling events with well-established roles in embryonic skin and hair follicle morphogenesis and systematically predicts a number of additional signaling patterns along dermal and epidermal cell lineage trajectories.

**CellChat predicts key signaling events between spatially colocalized cell populations.** To further demonstrate the predictive nature of CellChat, we studied signaling communication between E14.5 dermal condensate (DC) and epithelial placode cells, since these cells spatially colocalize and actively signal to each other during the initial stages of embryonic hair follicle formation (Fig. 4a). Three DC states—pre-DC, DC1, and DC2, and one placode state were identified (Supplementary Fig. 3e–f; see "Methods" section). CellChat analysis on these four cell states identified placode cells as the dominant communication "hub", which secretes and receives signals via 44 and 19 ligand-receptor pairs, respectively (Fig. 4b). Prominent bidirectional forward and reverse signals were identified for DC states, suggesting that the cell state transition along pre-DC-DC1-DC2 cell lineage trajectory is highly regulated. Specifically, FGF pathway exhibited abundant signaling interactions among all four states with FGF ligands being dominantly secreted by pre-DC and DC2 states (Fig. 4c). *Fgf10* was the major ligand contributing to dermal FGF signaling (Supplementary Fig. 5a), which is the known DC signature gene[36]. Epithelial placode cells distinctly secreted *Fgf20* both in autocrine and in paracrine manner to all three DC states (Supplementary Fig. 5a), which is consistent with the known role of placode-derived FGF20 signaling in hair follicle morphogenesis[22,42,43]. For another major signaling pathway in early hair follicle morphogenesis—canonical WNT, epithelial placode cells were the major source of ligands (Fig. 4c), prominently expressing primarily autocrine *Wnt3* and *Wnt6*. CellChat also predicted that this dominant epithelial autocrine WNT signaling was supplemented by a minor DC-derived *Wnt9a* paracrine signaling (Supplementary Fig. 5b-d). In contrast with canonical WNT, the inferred ncWNT signaling network revealed that DC cells express only one ligand, *Wnt5a*, that drives paracrine DC-to-placode and autocrine

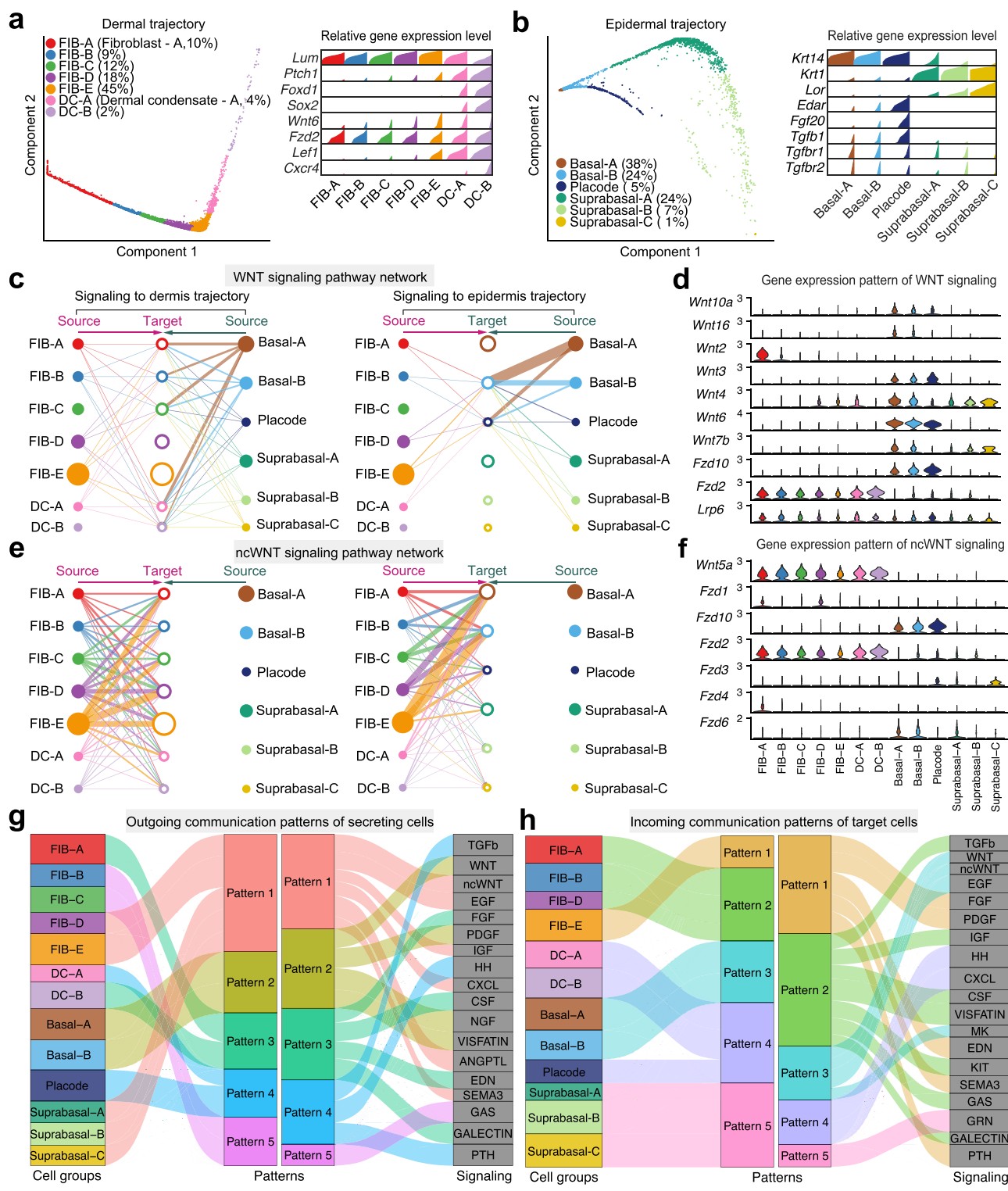

DC-to-DC signaling (Supplementary Fig. 5e). This result implies distinct roles for canonical WNT and ncWNT pathways in hair follicle morphogenesis.

By systematically investigating the predicted placode-to-DC signals, we found 21 ligand-receptor pairs implicating WNT, TGFβ, SEMA3, PTN, PDGF, MK, and FGF signaling pathways in the process of DC specification (Fig. 4d). Pattern recognition analysis further revealed that pre-DC and DC2 states jointly coordinate outgoing signals for ncWNT, FGF, IGF, EDN, and SEMA3 pathways (pattern #1 in Fig. 4e). DC1 dominantly drives

PROS signaling (pattern #3), while epithelial placode cells drive outgoing WNT, TGFβ, PDGF, MK, PTN, and PTH signaling (pattern #2, Supplementary Fig. 5f). At the incoming end of signaling, pre-DC cells respond to SEMA3 and PTH signaling (pattern #3 in Fig. 4f); DC1 and DC2 cells respond to TGFβ, PDGF, EDN, and PROS signaling (pattern #1) and epithelial placode cells respond to WNT, ncWNT, IGF, MK, and PTN signaling (pattern #2, Supplementary Fig. 5f).

CellChat revealed that at E14.5, DC cells respond to autocrine PROS signaling (Fig. 4g). *Pros1* is the ligand for the pathway,

**Fig. 3 Application of CellChat to continuous cell states along pseudotemporal trajectories during embryonic skin development. a** Left: Diffusion map projecting dermal skin cells onto the low-dimensional space and showing the dermal differentiation from fibroblasts to DC (dermal condensate) cells. Cells are grouped based on their location in this space. Right: Density plot showing the distribution of expression for selected marker genes in each cell group/ population. **b** Diffusion map showing the epidermal trajectory and associated density plot for selected marker genes. **c** Hierarchical plot showing dermal and epidermal interactions via canonical WNT signaling. Left and right portions show the autocrine and paracrine signaling to dermal trajectory and epidermal trajectory, respectively. Circle sizes are proportional to the number of cells in each cell group and edge width represents the communication probability. **d** Violin plot showing the expression distribution of signaling genes involved in the inferred WNT signaling network. **e** The dermal and epidermal interactions via ncWNT signaling. **f** The expression distribution of signaling genes involved in the inferred ncWNT signaling network. **g** The outgoing signaling patterns of secreting cells visualized by alluvial plot, which shows the correspondence between the inferred latent patterns and cell groups, as well as signaling pathways. The thickness of the flow indicates the contribution of the cell group or signaling pathway to each latent pattern. The height of each pattern is proportional to the number of its associated cell groups or signaling pathways. Outgoing patterns reveal how the sender cells coordinate with each other, as well as how they coordinate with certain signaling pathways to drive communication. **h** Incoming signaling patterns of target cells. Incoming patterns show how the target cells coordinate with each other, as well as how they coordinate with certain signaling pathways to respond to incoming signaling.

which signals via the receptor tyrosine kinase *Axl*. Signaling via *Axl* has been implicated in conferring cells with migratory properties in different biological context, including EMT-mediated cancer invasion[46–48], and directional migration has been recently shown to be crucial for normal dermal condensate formation upon hair follicle morphogenesis[42]. We examined CellChat's prediction of active PROS signaling in DC cells by RNAscope technique for *Edn3* as DC marker, *Axl* and *Thy1* (*Cd90*) as a marker of cell migration[49,50] and EMT process[51]. As expected from scRNA-seq, *Axl* expression was co-localized with *Edn3* and *Thy1* expression, which was concentrated in DC with significantly lower levels elsewhere (Fig. 4h). This RNAscope result is consistent with the possibility of autocrine PROS signaling in DC, likely driven via *Pros1-Axl* signaling.

During early hair follicle formation at E14.5, melanoblasts (melanocyte precursor cells) migrate into the hair placode from the dermis and then become differentiated toward melanocytes. However, the mechanisms of melanocyte migration into placode remain incompletely understood[52]. Therefore, we further studied the cell–cell communication among placodes, DC cells and melanocyte cells (including three melanocyte subpopulations: MELA-A, MELA-B, and MELA-C; see "Methods" section and Supplementary Fig. 3g). CellChat revealed that melanocytes strongly respond to DC cells via previously unrecognized EDN signaling (Fig. 4i). *Edn3* is a ligand of the EDN pathway, which regulates melanocyte migration[53]. Therefore, CellChat prediction suggests DC cells induce early directed migration of melanocytes. To experimentally examine this prediction, we used the RNAscope technique to spatially map expression of *Dct*, which marks late-stage melanocyte precursors, *Edn3* ligand and its receptor *Ednrb* in E14.5 embryonic mouse skin. As expected, *Dct*+ melanocytes (i.e., MELA-C subpopulation) localize in and around epithelial placode. They also express *Ednrb*. In turn, *Edn3* is specifically enriched in DC cells (preDC, DC1, and DC2 subpopulations), while *Ednrb* is also enriched in a portion of DC cells (likely DC2 subpopulation). Scattered *Ednrb*+/*Edn3*^neg/*Dct*^neg cells inside dermal condensate are likely undifferentiated melanoblasts (i.e., MELA-A/B subpopulations) (Fig. 4j). This spatial *Edn3*, *Ednrb*, *Dct* co-expression pattern is highly consistent with the scRNA-seq data (Fig. 4i). Thus, our RNAscope result confirms the CellChat prediction of *Edn3-Ednrb* signaling from DC cells to melanocytes, implying the roles for DC cells in inducing early-stage directed migration of melanocytes into placodes. It also shows potential autocrine *Edn3-Ednrb* signaling within the dermal condensate.

**Joint learning of time-course scRNA-seq data to uncover dynamic communication patterns.** Next, we demonstrate how CellChat can be applied to studying temporal changes of intercellular communications in the same tissue (Fig. 5a). For this

purpose, we performed combined analysis on two embryonic mouse skin scRNA-seq datasets from days E13.5 and E14.5[22]. Unsupervised clustering of E13.5 and E14.5 datasets identified 11 skin cell populations at E13.5 and E14.5 and additional two populations (i.e., dermal DC and pericytes) specific to E14.5 (Supplementary Fig. 3a–d; see "Methods" section).

We inferred intercellular communications for the above two datasets separately, and then analyzed them together via joint manifold learning and classification of the inferred communication networks based on their functional similarity. The functional similarity analysis requires the same cell population composition between two datasets. Thus, for such analysis we used only 11 common cell populations between E13.5 and E14.5 datasets. As the result, the signaling pathways associated with inferred networks from both datasets were mapped onto a shared two-dimensional manifold and clustered into groups. We identified four pathway groups (Fig. 5b-c). Groups #1 and #3 were dominated by growth factor pathways such as PDGF, NGF, FGF, EGF, and ANGPTL, while groups #2 and #5 dominantly contained inflammation-related pathways such as CCL, IL2, IL4, OSM, LIFR, and VISFATIN. As expected, the majority of the same signaling pathways from E13.5 and E14.5 were grouped together such as CCL, CSF, ANGPTL, PDGF, VEGF, ncWNT, and MK, suggesting that these pathways are essential for skin morphogenesis at both time points and likely do not critically regulate new developmental events at E14.5, such as hair follicle morphogenesis or dermal maturation. However, WNT and KIT signaling were classified into different groups, consistent with profound and multi-faceted role of WNT signaling in skin morphogenesis[22,54]. By computing the Euclidean distance between any pair of the shared signaling pathways in the shared two-dimensional manifold, we observed a large distance for WNT and KIT and to a lesser extent for RANKL, IL2, FGF, GALECTIN, EGF, TGFβ, and NGF pathways (Fig. 5d, Supplementary Fig. 6a-d). We specifically examined how WNT communications change over one day of skin development (Fig. 5e-j, Supplementary Data 2). At both embryonic time points, basal epidermal cells were the dominant source of WNT ligands, with further minor contribution from fibroblasts. Yet, compared to E13.5, when only basal epidermal cells were the WNT targets, at E14.5 fibroblasts gained WNT responsiveness. Further, melanocytes emerged as the new minor source of WNT signaling, helping to drive an overall increase in WNT communication network complexity. Collectively, the joint manifold learning enables the identification of signaling pathways that undergo embryonic stage-dependent change.

Next, we compared the information flow for each signaling pathway between E13.5 and E14.5 time points. The information flow for a given signaling pathway is defined by the sum of

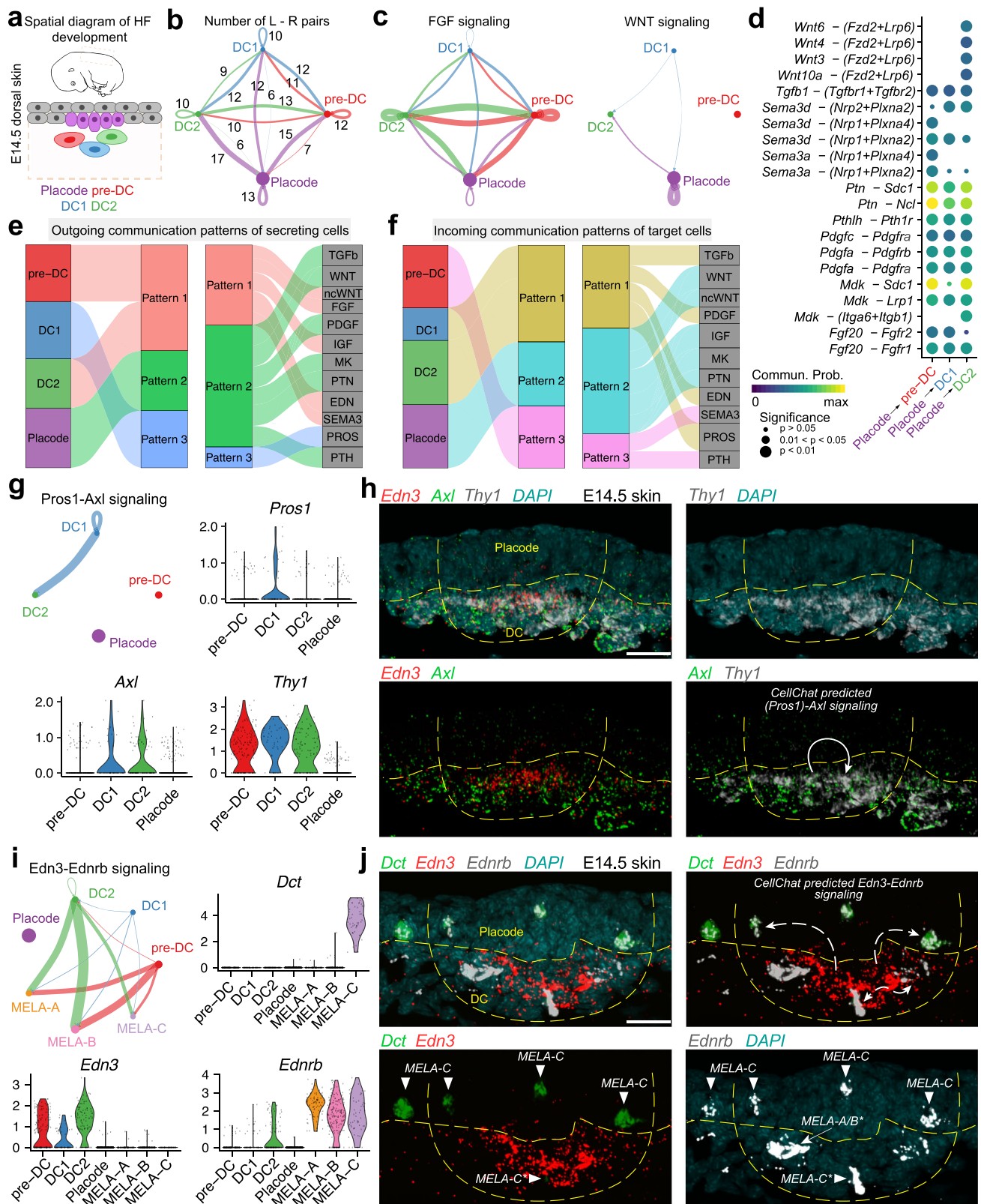

communication probability among all pairs of cell groups in the inferred network. We found that some pathways, including ANGPTL, APELIN, CSF, FGF, RANKL, and TGFβ maintain similar flow between the time points (black in Fig. 5k). We interpret that these pathways are equally important in the developing skin at both time points. In contrast, other pathways prominently change their information flow at E14.5 as compared

to E13.5: (i) turn off (NT, TWEAK), (ii) decrease (such as PTN, MK), (iii) turn on (TNF), or (iv) increase (such as WNT, GALECTIN, KIT, IGF, VEGF).

Moreover, we studied the detailed changes in the outgoing signaling across all significant pathways using pattern recognition analysis (Fig. 5l; see "Methods" section). We found that skin fibroblasts change their major and minor outgoing communication

**Fig. 4 Application of CellChat to communications between spatially colocalized cell populations. a** Spatial diagram of placode, pre-DC, DC1 and DC2 cells during hair follicle (HF) development at E14.5. DC: dermal condensate. **b** Number of significant ligand-receptor pairs between any pair of two cell populations. The edge width is proportional to the indicated number of ligand-receptor pairs. **c** The inferred FGF and WNT signaling networks. Circle sizes are proportional to the number of cells in each cell group and edge width represents the communication probability. **d** All the significant ligand-receptor pairs that contribute to the signaling sending from placode to three DC states. The dot color and size represent the calculated communication probability and *p*-values. *p*-values are computed from one-sided permutation test. **e** The outgoing communication patterns of secreting cells, which shows the correspondence between the inferred latent patterns and cell groups, as well as signaling pathways. **f** Incoming communication patterns of target cells. **g** The inferred Pros1-Axl signaling network, as well as the scRNA-seq expression distribution of the *Pros1* ligand, the *Axl* receptor and cell migration marker *Thy1*. The edge width represents the communication probability. **h** RNAscope data (*n* =4 independent experiments) showing spatial distribution of *Edn3* (*red*), *Axl* (*green*), and *Thy1* (*white*) transcripts in early-stage developing hair follicle from E14.5 embryonic mouse skin. Epithelial placode and dermal condensate (DC) are annotated and outlined with dashed lines. Solid white curved arrows in the bottom-right panel mark CellChat-predicted Pros1-Axl signaling within skin space. DAPI (teal) stains nuclei. Scale bar: 50 μm. **i** The inferred Edn3-Ednrb signaling network, as well as the scRNA-seq expression distribution of the melanocyte marker *Dct*, *Edn3* ligand and its receptor *Ednrb*. DC: dermal condensate; MELA: melanocytes; (**j**) RNAscope data (*n* = 4 independent experiments) showing spatial distribution of *Dct* (*green*), *Edn3* (*red*), and *Ednrb* (*white*) transcripts in early-stage developing hair follicle from E14.5 embryonic mouse skin. Arrowheads mark possible melanocyte populations. Solid white curved arrows in the top-right panel mark CellChat-predicted Edn3-Ednrb signaling within skin space. DAPI (teal) stains nuclei. Scale bar: 50 μm.

patterns between E13.5 and E14.5. At E13.5, early fibroblast state FIB-A dominates the outgoing signaling. Over one day period, the minor signaling of late fibroblast states FIB-B and FIB-P become major and includes ANGPTL, IGF, VEGF, KIT, SEMA3 pathways (Supplementary Fig. 6a–h). This suggests the balancing changes in the levels and patterns of ligand expression. On the other hand, endothelial cells (ENDO), melanocytes (MELA) and skin-resident myeloid cells (MYL) maintain their outgoing signaling patterns. Complex outgoing signaling dynamics were observed in the epidermis. Basal epidermal cells at E14.5 maintain secreted signaling patterns for NGF, PDGF, VISFATIN, and WNT, yet turn off signaling including for KIT and Neurotrophin (NT), and turn on signaling including for VEGF, PTN and LIFR. On the other hand, spinous epidermal cells prominently redesign their outgoing signaling. They turn off or decrease four pathways, such as PDGF (Supplementary Fig. 6e and 6g), turn on SEMA3 pathway, and maintain three pathways—IGF, MK, and PTN (Supplementary Fig. 6f and 6h). Prominent change in spinous cell signaling is consistent with known epidermal stratification event that occurs in mice at the transition between E13.5 and E14.5[55,56]. Taken together, CellChat analysis on joint scRNA-seq datasets enables multifaceted assessment of intercellular communication patterns across biological times, such as embryonic developmental time scale.

**Joint learning of conserved and context-specific communication patterns between distinct scRNA-seq datasets.** We also used CellChat to compare cell–cell communication patterns between two scRNA-seq datasets, one from embryonic day E13.5 skin[22] and another from adult day 12 wound skin[23] (Fig. 6a). While representing the same tissue (skin) from the same species (mouse) and containing some of the same principal cell types, such as fibroblasts, these two datasets are from vastly distinct biological contexts— embryonic morphogenesis vs. wound-induced repair. As such, this case study presents an opportunity to discover signaling logic and signal conservation principles. First, we performed joint manifold learning and classification of the inferred communication networks based on their topological similarity (functional similarity cannot be performed because of the vastly different cell type composition). We identified four signaling pathway groups (Fig. 6b–c). Intriguingly, none of the groups are unique to a given dataset, suggesting that the entire spectrum of communications is represented in both skin states. There are, however, dataset-specific enrichments, especially in groups #1 and # 4, which are dominated by signaling networks of the embryonic skin (8 out of 14 and 6 out of 9, respectively). The other two groups #2 and #3 are nearly equally contributed by the communication networks and contain several overlapping pathways from both skin states. By computing

the Euclidean distance between any pair of the shared signaling pathways in the shared two-dimensions space, we observed a large distance for signaling pathways like IGF, PDGF, CSF, PROS, and CCL (Supplementary Fig. 7a-b), suggesting that these pathways exhibit significantly different communication network architectures. However, other signaling pathways show relatively small distances, including ANGPTL, RANKL, TGFb, SEMA3, IL2, PTN, ncWNT, MK, EGF, APELN, and EDN (Supplementary Fig. 7c), which are also grouped together (Fig. 6c–d). This suggests similar communication network architectures for these overlapping pathways in both skin states. Closer look at the MK (Midkine) pathway (Fig. 6e–f) shows its high signaling redundancy (i.e., multiple signaling sources) and high target promiscuity (i. e. all cell groups can function as MK targets). The latter finding suggests that certain pathways have highly conserved signaling architecture (i.e., high degree of redundancy) which is largely independent of the specific cellular composition of the tissue.

We also compared the information flow (i.e., the overall communication probability) across the two skin datasets. Intriguingly, 19 out of 34 pathways are highly active, albeit at different levels, both in embryonic skin and in adult skin wounds (Fig. 6g). These likely represent core signaling pathways necessary for skin function independent of the specific point in the biological time scale (i.e., embryonic vs. adult). Nine pathways are active only in embryonic skin. These include such important pathways for skin morphogenesis as FGF[37,43,57–60] and WNT[22,36–40]. Four pathways are specifically active in wounded skin, including known regulators of wound-induced skin repair SPP1 (osteopontin)[61–63], MIF (macrophage migration inhibitory factor)[64–66] and IL1[67–69]. Taken together, this CellChat approach allows system-level classification and discovery of signaling communication network architecture principles.

**Joint learning of normal and diseased human skin to discover major signaling changes in response to disease.** As CellChatDB also includes curated ligand-receptor interactions of human, we next employed CellChat to detect the signaling changes between so-called lesional (diseased) and nonlesional (normal) skin from patients with atopic dermatitis (AD) using recently published human skin scRNA-seq dataset[70] (Fig. 7a). The original study revealed that lesional skin was enriched for chemokine signals (including *CCL19*) from inflammatory fibroblasts to inflammatory immune cells, including dendritic cells (DC) and T cells (TC). This was validated using immunofluorescence staining[70]. Therefore, we used CellChat to study the intercellular communication among fibroblasts (four subpopulations: *APOE* + FIB, *FBN1* + FIB, *COL11A* + FIB, and Inflam.FIB), DCs (four

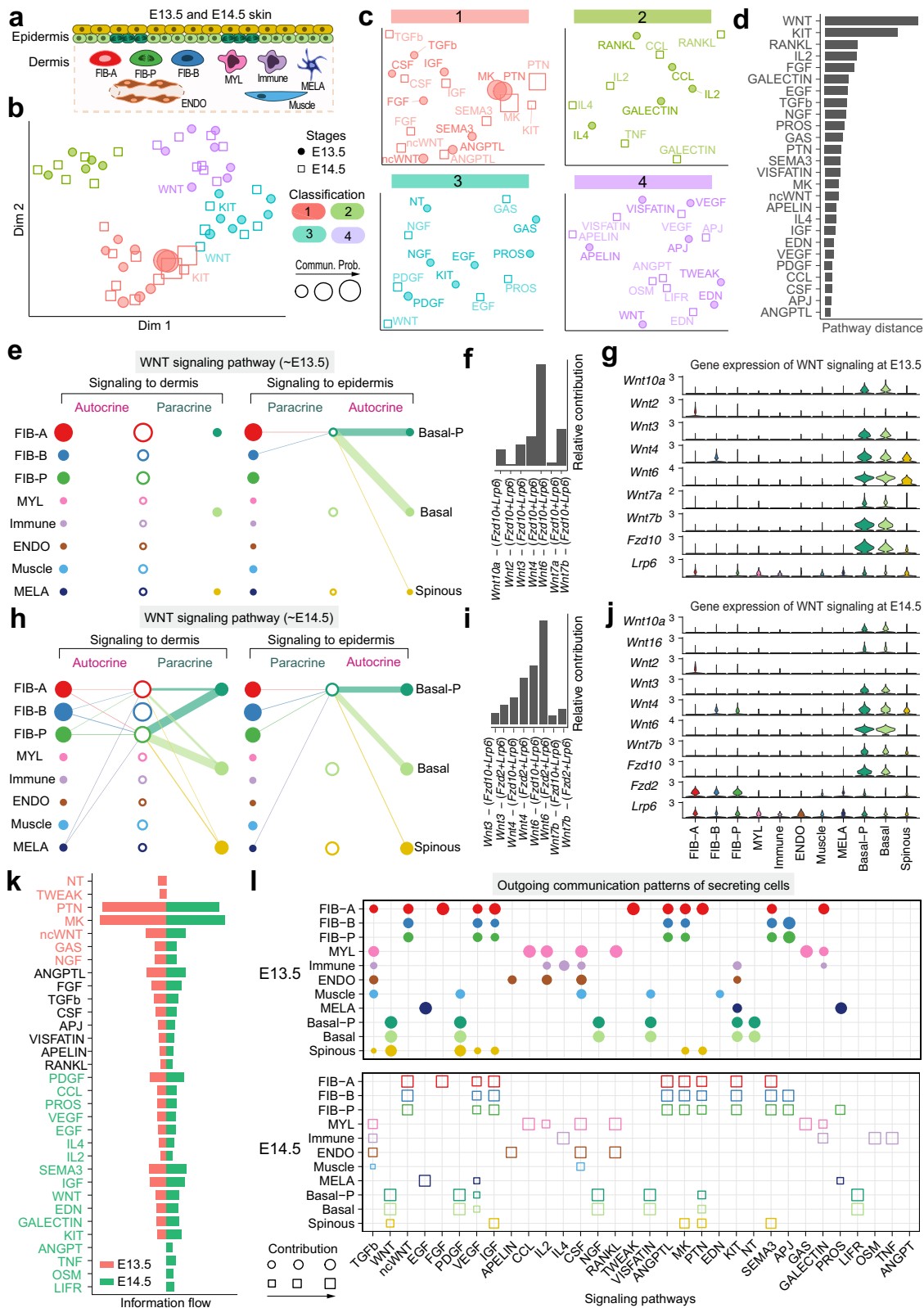

subpopulations: cDC1, cDC2, LC, and Inflam.DC), and TCs (four subpopulations: TC, Inflam.TC, *CD40LG* + TC and NKT) (Supplementary Fig. 8a–e, see "Methods" section).

We inferred intercellular communication networks for the nonlesional (NL) and lesional (LS) skin separately, and then jointly mapped them onto a shared two-dimensional manifold and clustered them into groups based on their functional similarity. We identified four pathway groups (Fig. 7b). Almost all the same signaling pathways from NL and LS were grouped together such as VEGF, GAS, LIGHT, CD40, and MIF, suggesting that these pathways are essential for both nonlesional and lesional skin and likely do not critically contribute to disease pathogenesis. By comparing the overall communication probability between nonlesional and lesional skin, we found that 11 out of 16 signaling

**Fig. 5 Comparison analysis of epidermal-dermal communications between different skin developmental stages. a** Schematic illustration of cellular composition of embryonic skin at E13.5 and E14.5. Different cell populations are color-coded to match colors in panel **e** and **h**. FIB-A: fibroblast type A; FIB-B: fibroblast type B; FIB-P: proliferative fibroblasts. MYL: myeloid cell; ENDO: endothelial cell; MELA: melanocytes; **b** Jointly projecting and clustering signaling pathways from E13.5 and E14.5 into a shared two-dimensional manifold according to their functional similarity. Circle and square symbols represent the signaling networks from E13.5 and E14.5 respectively. Each dot or square represents the communication network of one signaling pathway. Dot or square size is proportional to the total communication probability. Different colors represent different groups of signaling pathways. **c** Magnified view of each pathway group. **d** The overlapping signaling pathways between E13.5 and E14.5 were ranked based on their pairwise Euclidean distance in the shared two-dimensional manifold. **e** The inferred WNT signaling network at E13.5. Left and right portions show the autocrine and paracrine signaling to dermis and epidermis, respectively. Circle sizes are proportional to the number of cells in each cell group and edge width represents the communication probability. **f** Relative contribution of each ligand-receptor pair to the overall WNT signaling network at E13.5. **g** Expression distribution of WNT signaling genes at E13.5. **h** The inferred WNT signaling network at E14.5. **i** Relative contribution of each ligand-receptor pair at E14.5. **j** The expression distribution of WNT signaling genes at E14.5. **k** All significant signaling pathways were ranked based on their differences of overall information flow within the inferred networks between E13.5 and E14.5. The top signaling pathways colored red are more enriched in E13.5, the middle ones colored black are equally enriched in E13.5 and E14.5, and the bottom ones colored green are more enriched in E14.5. **l** The dot plot showing the comparison of outgoing signaling patterns of secreting cells between E13.5 and E14.5. The dot size is proportional to the contribution score computed from pattern recognition analysis. Higher contribution score implies the signaling pathway is more enriched in the corresponding cell group.

pathways were highly active in lesional skin, including 9 pathways involved in inflammatory and immune response, such as CXCL, LIGHT, GLAECTIN, COMPLEMENT, MIF, CSF, IL4, CCL, and TNF (Fig. 7c). Four pathways were specifically active in lesional skin, including known inflammatory signals CSF, IL4, CCL, and TNF, suggesting that these pathways might critically contribute to disease progression. Specific to CCL signaling, CellChat identified ligand-receptor pair *CCL19-CCR7* as the most significant signaling, contributing to the communication from Inflam.FIB to Inflam.DC (Fig. 7d–f). This is in agreement with a reported experimental finding[70]. Ligand *MIF* and its multi-subunit receptor *CD74/CD44* were found to act as major signaling from Inflam.FIB to Inflam.TC in lesional skin compared to nonlesional skin (Fig. 7d and Supplementary Fig. 9a–c). Ligand *CXCL12* and its receptor *CXCR4* were also found to be highly active in lesional skin, in particular, for the signaling from Inflam.FIB to cDC2 and Inflam.DC (Fig. 7d and Supplementary Fig. 9a–c). Together, CellChat's joint analysis using an example of human lesional and nonlesional skin enables the discovery of major signaling changes that might drive disease pathogenesis.

**Comparison with other cell–cell communication inference tools.** We compared CellChat with three other tools for inferring intercellular communications—SingleCellSignalR[9], iTALK[10], and CellPhoneDB[16] using the same four mouse skin datasets analyzed by CellChat (see "Methods" section). Currently existing tools, such as SingleCellSignalR and iTALK, typically use only one ligand/one receptor gene pairs, largely neglecting the effect of multiple receptors. We computed the percentage of false positive interactions caused by the above fact. False positive interactions are defined as the interactions with multi-subunits that are partially identified by these tools (see "Methods" section). We found that the average rate of false positive interactions identified by SingleCellSignalR and iTALK was 10.6% and 14.3%, respectively (Supplementary Fig. 10), suggesting the importance of accurate representation of known ligand-receptor interactions. Of note, failed detection of interactions with multi-subunits might be also caused by low expression of multi-subunits of the receptors that are not captured using scRNA-seq.

We also compared the performance of CellChat with CellPhoneDB, which considers multi-subunit ligand–receptor complexes. We reasoned that any given method can be regarded as more accurate if its predictions more significantly overlap with the predictions of more than one other method. We found that CellChat predictions had more overlapping interactions with both SingleCellSignalR and iTALK predictions across all four scRNA-seq datasets (Supplementary Fig. 11a). CellChat and CellPhoneDB

shared ~50% predicted interactions (Supplementary Fig. 11a). To assess the sensitivity of inferred communications to the input data, we used subsampling of 90, 80, or 70% of the total number of cells in each dataset, and then computed the true positive rate (TPR), false positive rate (FPR), and accuracy (ACC) by comparing subsampled datasets with the original dataset. CellChat produced a slightly higher TPR, lower FPR and higher ACC in comparison with CellPhoneDB (Supplementary Fig. 11b). Both CellChat and CellPhoneDB were relatively robust to subsampling, which is likely because both methods infer cell–cell communication based on cell clusters. Such robustness in terms of subsampling is very useful when analyzing the rapidly growing volume of scRNA-seq data.

Next, we compared cell–cell communication networks inferred by CellChat, CellPhoneDB, iTALK, and SingleCellSignalR using an example of four spatially colocalized cell populations in E14.5 embryonic mouse skin (Fig. 4). We compared the inferred significant ligand-receptor (L-R) pairs for any two cell subpopulations between CellChat and other methods. Here we only retained the top 10% of L-R pairs (the most significant) inferred by iTALK and SingleCellSignalR to ensure the comparable number of L-R pairs with that by CellChat. The average numbers of L-R pairs between two cell subpopulations inferred by the above four methods were 12, 37, 14, and 12, respectively (Supplementary Table 1). We found that CellChat shared more L-R pairs with CellPhoneDB than with iTALK, likely due to the fact that both CellChat and CellPhoneDB consider multi-subunit complexes and determine the significant L-R pairs using a statistical approach. SingleCellSignalR shared very few L-R pairs with the other three methods, suggesting a potentially different logic for quantifying and ranking L-R interactions. Moreover, the majority of shared L-R pairs between CellChat and CellPhoneDB were independently ranked as top pairs by CellPhoneDB (Supplementary Data 1). This result suggests that although CellChat infers fewer L-R pairs than CellPhoneDB, it captures the strongest (and likely the most significant) L-R interactions.

We also systematically evaluated different methods based on the assumption that spatially adjacent cell types should have stronger cell–cell communication than spatially distant cells. We have studied cell–cell communication for four spatially colocalized cell populations in E14.5 embryonic mouse skin, including Placodes, pre-DC, DC1, and DC2 (Fig. 4). We now added seven cell types that are likely not spatially adjacent to the above four cell populations–FIB (fibroblasts), MELA (melanocytes), Spinous (spinous epithelial cells), MYL (myeloid cells), Immune (other immune cells), ENDO (endothelial cells) and Muscle. We then computed the number of inferred interactions, as well as the sum

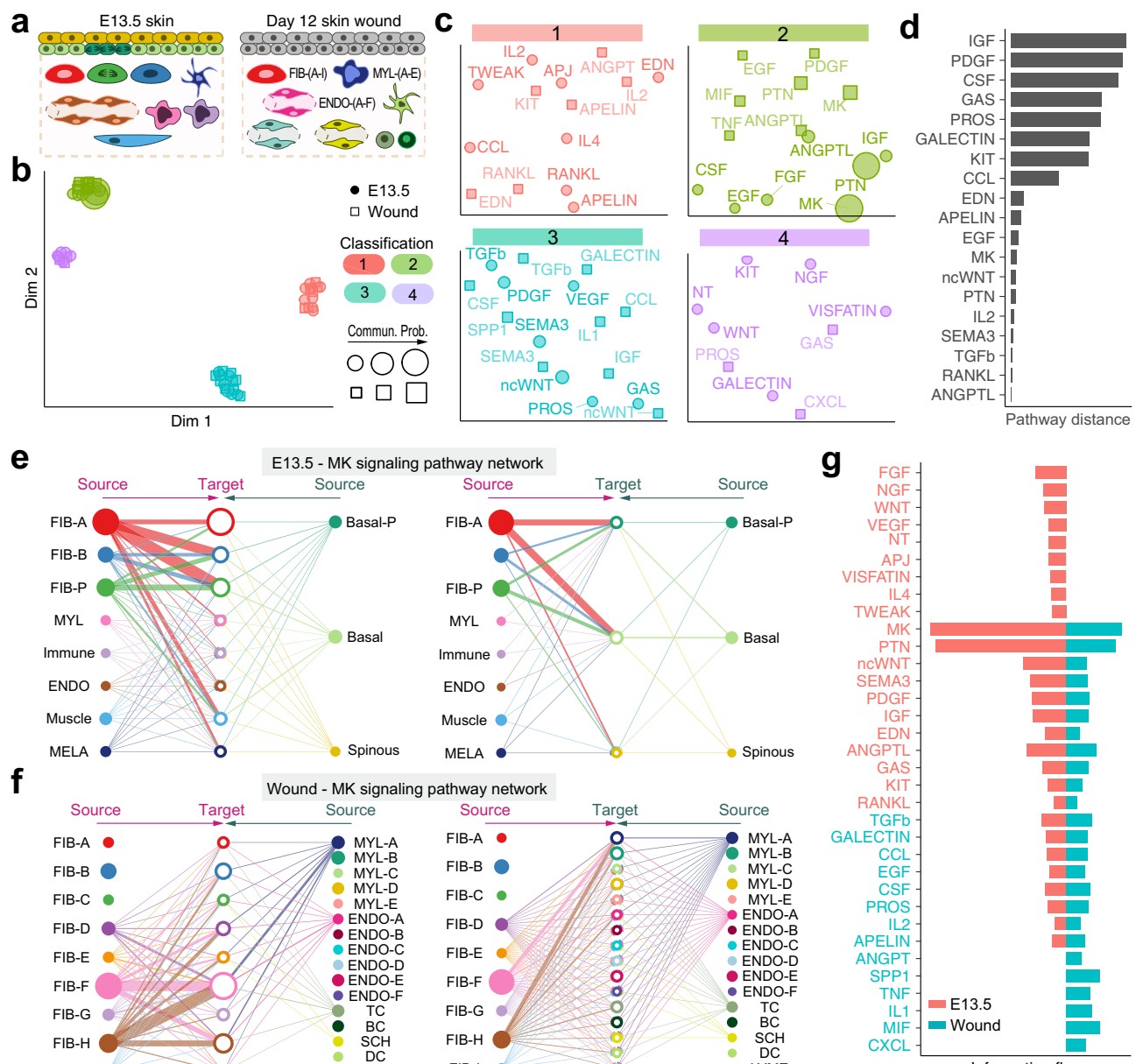

**Fig. 6 Joint identification of conserved and context-specific communication patterns between two skin states. a** Schematic illustration of cellular composition of skin during embryonic morphogenesis at E13.5 and during adult wound-induced repair at day 12. Different cell populations are color-coded to match colors in panel **e** and **f**, respectively. **b** Jointly projecting and clustering signaling pathways from E13.5 and wound onto shared two-dimensional manifold according to their structural similarity of the inferred networks. Circle and square symbols represent the signaling networks from E13.5 and wound respectively. Each circle or square represents the communication network of one signaling pathway. Circle or square size is proportional to the total communication probability of that signaling network. Different colors represent different groups of signaling pathways. **c** Magnified view of each pathway group. **d** The overlapping signaling pathways between E13.5 and wound were ranked based on their pairwise Euclidean distance in the shared two-dimensional manifold. Larger distance implies larger difference. **e–f** Hierarchical plot showing the inferred intercellular communication network of MK signaling pathway at E13.5 and wound, respectively. Circle sizes are proportional to the number of cells in each cell group and edge width represents the communication probability. **g** All the significant signaling pathways were ranked based on their differences of overall information flow within the inferred networks between E13.5 and wound. The overall information flow of a signaling network is calculated by summarizing all the communication probabilities in that network. The top signaling pathways colored by red are more enriched in E13.5, and the bottom ones colored by green were more enriched in the wound.

of interaction probabilities or scores between each cell type and the four spatially colocalized cell populations. We found that CellChat consistently captures stronger interactions in spatially adjacent cells than distant cells both in terms of the number of interactions and the interaction probabilities (Supplementary Fig. 12a–b). CellPhoneDB also performed well at discriminating spatially-adjacent from distant cells. iTALK failed to capture

stronger interactions in spatially adjacent cells as compared to spatially distant cells for FIB, MELA, MYL, and ENDO. SingleCellSignalR also failed for FIB and ENDO. By considering all seven cell types together, we found that both CellChat and CellPhoneDB can significantly distinguish spatially adjacent from distant cells, whereas iTALK and SingleCellSignalR predicted stronger interactions in spatially adjacent cells than distant cells

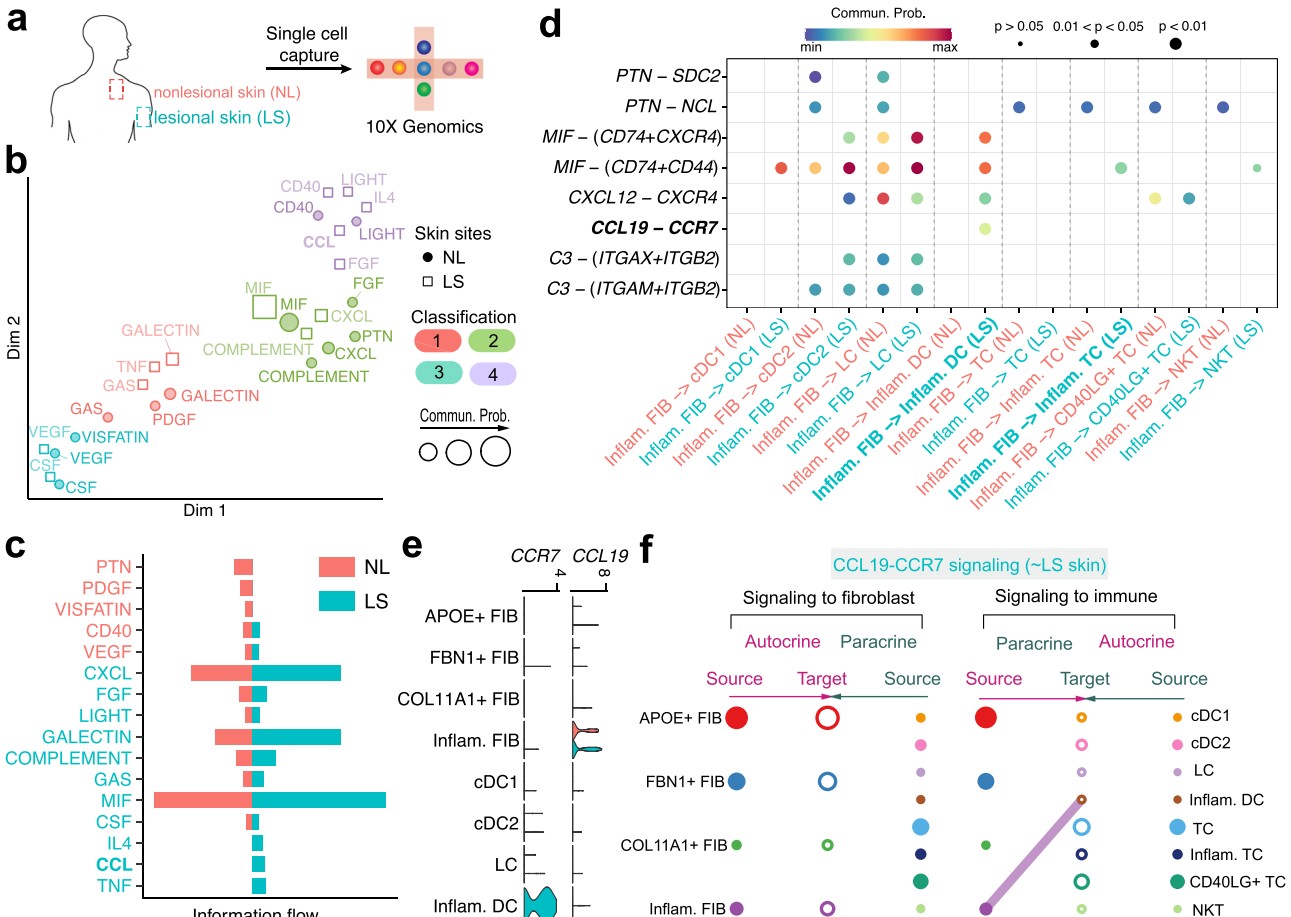

**Fig. 7 Identification of major signaling changes in diseased human skin. a** Schematic illustration of scRNA-seq on cells from nonlesional (NL, normal) and lesional (LS, diseased) human skin from patients with atopic dermatitis. **b** Jointly projecting and clustering signaling pathways from NL and LS skin onto shared two-dimensional manifold according to functional similarity of the inferred networks. Circle and square symbols represent the signaling networks from NL and LS respectively. Each dot or square represents the communication network of one signaling pathway. Dot or square size is proportional to the communication probability. Different colors represent different groups of signaling pathways. **c** Significant signaling pathways were ranked based on differences in the overall information flow within the inferred networks between NL and LS skin. The overall information flow of a signaling network is calculated by summarizing all communication probabilities in that network. The top signaling pathways colored red are enriched in NL skin, and these colored green were enriched in the LS skin. **d** Comparison of the significant ligand-receptor pairs between NL and LS skin, which contribute to the signaling from Inflam.FIB (i.e., inflammatory fibroblasts) to dendritic cells (DC) and T cells (TC) including cDC1, cDC2, LS, Inflam.DC, TC, Inflam.TC, *CD40LG* + TC, and NKT subpopulations. The highlighted *CCL19-CCR7* signaling was previously validated using immunofluorescence staining. Dot color reflects communication probabilities and dot size represents computed p-values. Empty space means the communication probability is zero. p-values are computed from one-sided permutation test. **e** Expression distribution of ligand *CCL19* and its receptor *CCR7* in NL (red) and LS (green) skin. **f** Hierarchical plot showing inferred intercellular communication network of *CCL19-CCR7* signaling in LS skin. Left and right portions show autocrine and paracrine signaling to fibroblast and immune cells, respectively. Circle sizes are proportional to the number of cells in each cell group and edge width represents the communication probability. Note that CellChat predicted no significant *CCL19-CCR7* signaling in NL skin. FIB: fibroblasts; Inflam.FIB: inflammatory fibroblasts; cDC: conventional dendritic cell; Inflam.DC: inflammatory dendritic cell; LC: Langerhans cell; TC: T cell; Inflam.TC: inflammatory T cell; NKT: natural killer T cell.

with no statistically significant differences (Supplementary Fig. 12c). Since CellPhoneDB infers more interactions than CellChat, we tested whether the top interactions predicted by CellPhoneDB can also distinguish spatially adjacent from distant cells. For the top 10%, top 20% and top 30% interactions predicted by CellPhoneDB, the difference between spatially adjacent and distant cells was not as significant as with CellChat (Supplementary Fig. 13a–b), suggesting that CellChat performed better at capturing stronger interactions. Together, our analyses show that although CellChat produces fewer interactions, it performs well at predicting stronger interactions.

The unique characteristics and capabilities of CellChat and its comparison with other relevant tools are summarized in Supplementary Table 2. First, CellChatDB database incorporates

not only multi-subunit structure of ligand-receptor complexes but also soluble and membrane-bound stimulatory and inhibitory cofactors, leading to a more comprehensive database than those used by other tools. We also quantitatively showed the differences and the strengths of CellChatDB in comparison to other existing analogous databases, including CellTalkDB[71], CellPhoneDB[16], iTALK[10], SingleCellSignalR[9], Ramilowski2015[72], NicheNet[13], and ICELLNET[73]. Compared to the above databases, CellChatDB provides an important resource for the community to study biologically meaningful cell–cell communication (Supplementary Fig. 1b and Supplementary Note 1). Second, CellChat allows users to input a low-dimensional representation of the data, a particularly useful function when analyzing continuous states along pseudotime trajectories. Third, CellChat can extract higher

order information from the inferred communications for identification of major signaling sources, targets and essential mediators, as well as the prediction of coordinated responses among different cell types. Fourth, CellChat can group signaling pathways based on similarity of their communication patterns to identify signaling pathways with similar architectures, and possibly functions. Finally, CellChat can uncover conserved vs. context-specific communication patterns through manifold learning of multiple communication networks simultaneously.

## Discussion

In this work we report a database of signaling ligand-receptor interactions that considers the multimeric structure of ligand-receptor complexes and additional effects on the core interaction by soluble and membrane-bound stimulatory and inhibitory cofactors. The ligand-receptor pairs are also classified into functionally related signaling pathways via systematic manual curation based on peer-reviewed literature. Comprehensive recapitulation of known molecular interactions is essential for developing biologically meaningful understanding of intercellular communications from scRNA-seq data. For example, signaling via BMP, IL, Interferon, TGFβ pathways requires the presence of more than one membrane-bound receptor subunits. Further, many pathways, such as BMP and WNT, are prominently modulated by their cofactors, both positively and negatively. To our knowledge, CellChatDB is the first manually curated signaling interaction database in mouse that considers multimeric structure. Although users can map human genes to their mouse orthologues using available tools such as biomaRt[74], some molecular interactions are found in mouse but not in human and vice-versa and these are typically lost during such mapping. Cell-ChatDB additionally provides the signaling interactions in human by first automatically mapping to human orthologues and then manually adding the interactions specific to human.

Integration of all known molecular interactions when studying intercellular communication requires new modeling frameworks. To this end, we derived a mass action-based model for quantifying the communication probability between a given ligand and its cognate receptor. We modeled the signaling communication probability between two cell groups by considering the proportion of cells in each group across all sequenced cells. This is important because abundant cell populations tend to send collectively stronger signals than the rare cell populations. With the increasing number of datasets on unsorted single-cell transcriptomes in the Human Cell Atlas, tools with such consideration will be potentially in high demand. For the users who are interested in analyzing sorting-enriched single cells, we provide an option of removing the potential artifact of population size when inferring cell–cell communication. In addition, CellChat estimates the level of ligands by the geometric mean of the subunits. Due to the low amounts of mRNA in individual cells, dropout events often occur in scRNA-seq data[75], leading to possible zero expression of sub-units. However, dropouts are unlikely to affect strong signals predicted by CellChat because dropouts commonly happen for genes with low expression[75,76].

CellChat R package is a versatile and easy-to-use toolkit for inferring, analyzing and visualizing cell–cell communication from any given scRNA-seq data. It provides several graphical outputs to facilitate different post-analysis tasks. Of particular note is our customized hierarchical plot that provides an intuitive way to visualize oftentimes complex details of signaling by a given pathway, including: (i) clear view of source and target cell populations, (ii) easy-to-identified directionality and probability of signaling, and (iii) paracrine vs. autocrine signaling links. We demonstrated CellChat's diverse functionalities by applying it to finding continuous cell lineage-associated signaling events,

communications between spatially colocalized cell populations, temporal changes in time-course scRNA-seq data, and conserved and context-specific communications between datasets from distinct biological contexts.

A user-friendly web-based CellChat Explorer (http://www.cellchat.org/) was also built, which contains two major components: (a) Ligand-Receptor Interaction Explorer, which allows easy exploration of our ligand-receptor interaction database CellChatDB, and (b) Cell–Cell Communication Atlas Explorer, which allows easy exploration of the cell–cell communication. For any given scRNA-seq dataset that has been processed by our CellChat R-package, we can host the predicted results on our server, allowing easy exploration and comparison of cell–cell communication. While at present the Cell–Cell Communication Atlas only hosts the skin scRNA-seq datasets analyzed in this study, we envision its rapid growth to become a community-driven web portal for cell–cell communication in a broad range of tissues at single-cell resolution.

The successful performance of CellChat lies in utilizing a mass action-based model to integrate all known molecular interactions, including the core interaction between ligands and receptors with multi-subunit structure, and additional modulation by cofactors. While ligand-receptor interactions and law-of-mass-action happen at the protein level, mRNA levels are commonly used to approximate the protein level. A higher level of molecular details (e.g., protein levels in individual cells) could further improve the modeling accuracy of CellChat and related tools. Due to the technical difficulties of capturing single-cell proteomic information at present time, a comprehensive modeling of ligand-receptor interactions remains challenging. Determining a set of biologically meaningful parameters in the mass action model remains challenging, particularly when considering that different pairs of ligands and receptors often have different dissociation constants (i.e., the parameter $Kh$ in Hill function) and different degree of cooperativity (i.e., the parameter $n$ in Hill function). Although these parameters lack explicit biological connections in our current model, the Hill function can be considered as a nonlinear approximation of the ligand-receptor interactions. By computing the Jaccard similarity between the interactions inferred using different choices of the parameters $Kh$ and $n$, we noticed that the inferred ligand-receptor interactions by CellChat are relatively robust to those parameters within certain ranges for all four tested datasets (Supplementary Fig. 14).

CellChat communication pattern analysis can uncover coordinated responses among different cell types. Different cell types may simultaneously activate the same cell type-independent signaling patterns or different cell type-specific signaling patterns. Different numbers of patterns provide different resolutions when recovering coordinated responses (Supplementary Note 2; Supplementary Fig. 15). This analysis can potentially help to derive general cell–cell communication principles.

Cell clustering is a pre-requisite for cell–cell communication analysis with CellChat and other tools, such as CellPhoneDB, iTALK and SingleCellSignalR. While different number of cell clusters may naturally affect the inferred ligand-receptor interactions, with a fixed cluster number the clustering results using different methods or parameters will unlikely have major impact on the inferred ligand-receptor interactions. This is because our cell–cell communication is inferred at the cluster level, only depending on estimation of the average gene expression in each cell cluster. We demonstrated these two points using an example of E14.5 mouse embryonic skin dataset with four spatially colocalized cell subpopulations (Supplementary Note 2; Supplementary Fig. 16). In general, cell clustering needs to be carried out carefully in order to capture biologically meaningful cell populations before cell–cell communication analysis.

The number of inferred ligand-receptor pairs clearly depends on the method for calculating the average gene expression per cell group. Here we systematically explored the inferred ligand-receptor pairs using different methods by calculating the average gene expression per cell group, including mean (i.e., simply calculating the average gene expression), 5% truncated mean (i.e., calculating the average gene expression by discarding 5% from each end of the data), 10% truncated mean, trimean (i.e., the method used in CellChat) and median. For the four studied datasets, there are about 15% more dropped ligand-receptor pairs when calculating the average gene expression using trimean compared to the 10% truncated mean (Supplementary Fig. 17). Compared to other cell–cell communication tools, such as CellPhoneDB, which uses a 10% truncated mean, CellChat produces fewer ligand-receptor interactions. However, as seen in our comparison study on spatially adjacent subpopulations (Supplementary Fig. 13a, b), CellChat performs well at predicting stronger interactions.

Although we found CellChat's predictions can recapitulate known biology to a substantial degree, systematic evaluation of predicted cell–cell communication networks is challenging due to the lack of ground truth[7]. Here we employed three strategies to compare the performance of different computational methods. First, we reason that a more accurate method will have a larger proportion of overlapped predictions with other methods. However, such assumption has the following two limitations: (1) Similar methods tend to generate similar results regardless of accuracy; and (2) Different ligand-receptor databases used in each method could contribute to the variety of predicted interactions. Second, we comprehensively compared the inferred interactions for any two cell subpopulations on a specific dataset. We found that the shared interactions between CellChat and other methods were independently ranked as top pairs by other methods including CellPhoneDB. Third, we reason that spatially adjacent cell types should have stronger cell–cell communication than spatially distant cells. CellChat performs better in distinguishing spatially adjacent from distant cells both in terms of the number of interactions and the interaction strengths. Together, our analyses show that although CellChat produces fewer interactions than other methods, it performs well at predicting stronger interactions, which is helpful for narrowing down on interactions for further experimental validations. Other types of single-cell data such as proteomics[77] and spatial transcriptomics[78] when available are also helpful and important to benchmark and optimize these cell–cell communication methods in future studies.

Recent advances in spatially resolved transcriptomic techniques offer an opportunity to explore spatial organization of cells in tissues[78]. The integration of spatial information with scRNA-seq data will likely offer new insights into cellular crosstalk[79,80]. The present version of CellChat provides an easy-to-use tool for intercellular communication analysis on conventional, non-spatially resolved scRNA-seq data. While it remains to be tested, we believe it can be relatively easily adjusted, such as via introduction of spatial constrains on cell–cell signaling, to build intercellular communication networks on spatially resolved transcriptomic datasets. As single-cell multi-omics data is becoming more common[81,82], we anticipate that methods like CellChat, which are able to perform system-level analyses, will serve as useful hypothesis-generating tools whose predictive power will extend beyond the ability to classify cell populations and establish their lineage relationships, which currently dominate single-cell genomics studies.

## Methods

**Database construction for ligand-receptor interactions**. To construct a database of ligand-receptor interactions that comprehensively represents the current state of knowledge, we manually reviewed other publicly available signaling pathway databases, as well as peer-reviewed literature and developed CellChatDB. Cell-ChatDB is a database of literature-supported ligand-receptor interactions in both mouse and human. The majority of ligand–receptor interactions in CellChatDB were manually curated on the basis of KEGG (Kyoto Encyclopedia of Genes and Genomes) signaling pathway database (https://www.genome.jp/kegg/pathway.html). Additional signaling molecular interactions were gathered from recent peer-reviewed experimental studies. We took into account not only the structural composition of ligand-receptor interactions, that often involve multimeric receptors, but also cofactor molecules, including soluble agonists and antagonists, as well as co-stimulatory and co-inhibitory membrane-bound receptors that can prominently modulate ligand-receptor mediated signaling events. The detailed steps for how CellChatDB was built and how to update CellChatDB by adding user-defined ligand-receptor pairs were provided in Supplementary Note 1. To further analyze cell–cell communication in a more biologically meaningful way, we grouped all of the interactions into 229 signaling pathway families, such as WNT, ncWNT, TGFβ, BMP, Nodal, Activin, EGF, NRG, TGFα, FGF, PDGF, VEGF, IGF, chemokine and cytokine signaling pathways (CCL, CXCL, CX3C, XC, IL, IFN), Notch and TNF. The supportive evidences for each signaling interaction is included within the database.

**Inference of intercellular communications**. a) Identification of differentially expressed signaling genes. To infer the cell state-specific communications, we first identified differentially expressed signaling genes across all cell groups within a given scRNA-seq dataset, using the Wilcoxon rank sum test with the significance level of 0.05.

b) Calculation of ensemble average expression. To account for the noise effects, we calculated the ensemble average expression of signaling genes in a given cell group using a statistically robust mean method:

$$\text{EM} = \frac{1}{2}Q_2 + \frac{1}{4}(Q_1 + Q_3) \tag{1}$$

where $Q_1$, $Q_2$, and $Q_3$ is the first, second and third quartile of the expression levels of a signaling gene in a cell group.

c) Calculation of intercellular communication probability. We modeled ligand-receptor mediated signaling interactions using the law of mass action. Since the physical process of ligand-receptor binding involves protein-protein interactions, we used a random walk based network propagation technique[83,84] to project the gene expression profiles onto a high-confidence experimentally validated protein-protein network from STRINGdb[83,85]. Based on the projected ligand and receptor profiles, the communication probability $P_{i,j}$ from cell groups $i$ to $j$ for a particular ligand-receptor pair $k$ was modeled by:

$$
\begin{aligned}
P_{i,j}^k = {} & \frac{L_i R_j}{K_h + L_i R_j} \times \left(1 + \frac{AG_i}{K_h + AG_i}\right) \cdot \left(1 + \frac{AG_j}{K_h + AG_j}\right) \\
& \times \frac{K_h}{K_h + AN_i} \cdot \frac{K_h}{K_h + AN_j} \times \frac{n_i n_j}{n^2}, \\
& L_i = \sqrt[m1]{L_{i,1} \cdots L_{i,m1}}, \quad R_j = \sqrt[m2]{R_{j,1} \cdots R_{j,m2}} \cdot \frac{1 + RA_j}{1 + RI_j}.
\end{aligned}
\tag{2}
$$

Here $L_i$ and $R_j$ represent the expression level of ligand $L$ and receptor $R$ in cell group $i$ and cell group $j$, respectively. The expression level of ligand $L$ with $m1$ subunits (i.e., $L_{i,1}, \cdots, L_{i,m1}$) was approximated by their geometric mean, implying that the zero expression of any subunit leads to an inactive ligand. Similarly, we computed the expression level of receptor $R$ with $m2$ subunits. In addition, co-stimulatory and co-inhibitory membrane-bound receptors are capable of modulating signaling via the control of receptor activation[86]. For the ligand-receptor pair with multiple co-stimulatory receptors, we computed the average expression of these co-stimulatory receptors (denoted by RA) and then used a linear function to model the positive modulation of the receptor expression. For each ligand-receptor pair with multiple co-inhibitory receptors, we modeled them using the same approach. A Hill function was used to model the interactions between $L$ and $R$ with a parameter $K_h$ whose default value was set to be 0.5 as the input data has a normalized range from 0 to 1. The extracellular agonists and antagonists from both sender and receiver cells are able to directly or indirectly modulate the ligand-receptor interaction[86]. For the ligand-receptor pair with multiple soluble agonists, we computed the average expression of these agonists (denoted by AG) and then used a Hill function to model the positive modulation of the ligand-receptor interaction. For the ligand-receptor pair with multiple soluble antagonists, we modeled them using the same approach. The effect of cell proportion in each cell group was also included in the probability calculation when analyzing unsorted single-cell transcriptomes, where $n_i$ and $n_j$ are the numbers of cells in cell groups $i$ and $j$, respectively, and $n$ is the total number of cells in a given dataset. Together, the communication probabilities among all pairs of cell groups across all pairs of ligand-receptor were represented by a three-dimensional array $\mathbf{P}$ ($K \times K \times N$), where $K$ is the number of cell groups and $N$ is the number of ligand-receptor pairs or signaling pathways. The communication probability of a signaling pathway was computed by summarizing the probabilities of its associated ligand-receptor pairs. It should be noted that we did not perform normalization along the second dimension of $\mathbf{P}$ such that $\sum_j P_{i,j}^k = 1$ because the normalized data are not

suitable for comparing the communication probability between different cell groups across multiple signaling pathways. The communication probability here only represents the interaction strength and is not exactly a probability.

d) Identification of statistically significant intercellular communications. The significant interactions between two cell groups are identified using a permutation test by randomly permuting the group labels of cells, and then recalculating the communication probability $P_{i,j}$ between cell group $i$ and cell group $j$ through a pair of ligand $L$ and receptor $R$. The $p$-value of each $P_{i,j}$ is computed by:

$$p\text{-}value = \frac{\left\{ \#m | P_{i,j}^{(m)} \le P_{i,j}, m = 1, 2, \cdots, M \right\}}{M} \quad (3)$$

where the probability $P_{i,j}^{(m)}$ is the communication probability for the $m$-th permutation. $M$ is the total number of permutations ($M = 100$ by default). The interactions with $p$-value <0.05 are considered significant.

### Discovery of dominant senders, receivers, mediators, and influencers in the intercellular communication networks.
To allow ready identification of major signaling sources, targets, essential mediators and key influencers, as well as other high-order information in intercellular communications, the centrality metrics from graph theory, previously used for social network analysis, were adopted[17]. Specifically, we used measures in weighted-directed networks, including out-degree, in-degree, flow betweenesss and information centrality, to respectively identify dominant senders, receivers, mediators and influencers for the intercellular communications. In a weighted-directed network with the weights as the computed communication probabilities, the out-degree, computed as the sum of communication probabilities of the outgoing signaling from a cell group, and the in-degree, computed as the sum of the communication probabilities of the incoming signaling to a cell group, can be used to identify the dominant cell senders and receivers of signaling networks, respectively. Flow betweenness score[87] measures a group of cells' capability as gatekeeper to control communication flow between any two cell groups. Information centrality score provides a hybrid measure, for example by combining closeness and eigenvector, for information flow within a signaling network, and a higher value indicates greater control on the information flow[87]. Other popular centrality metrics, such as hub, authority, EigenCentrality and PageRank[88], can be also used to identify highly influential cell groups in the intercellular communications. The flow betweenness and information centrality are calculated by the package sna[87]. Other measures are computed by the package igraph (https://igraph.org/).

### Identification of major signals for specific cell groups and global communication patterns.
To identify key signals and latent communication patterns among all signaling pathways, CellChat uses an unsupervised learning method non-negative matrix factorization that has been successfully applied in pattern recognition[18,19,82,89]. First, the latent patterns were found for sending cells by summarizing the communication probability array $\mathbf{P}$ (three-dimensional) along the second dimension to obtain a two-dimensional matrix $\mathbf{P_j}$. A non-negative matrix factorization was then carried out via:

$$\min_{W,H>0} \left\| P_j - WH \right\|, \quad (4)$$

where the two low-dimensional matrices $\mathbf{W}$ and $\mathbf{H}$ are the cell loading and signaling loading matrices with sizes $K \times R$ and $R \times N$, respectively. Each of the $R$ columns in $\mathbf{W}$ and the corresponding rows in $\mathbf{H}$ is considered as a communication pattern. $W_{ir}$ is the loading values of cell group $i$ in pattern $r$, representing the contributions of cell group $i$ in pattern $r$. $H_{rk}$ represents the contributions of ligand-receptor pair or signaling pathway $k$ in pattern $r$. As the number of patterns increases, there might be redundant patterns, making it difficult to interpret the communication patterns. We chose five patterns as the initial default because the number of cell groups and significant signaling pathways are relatively small. In addition, we inferred the number of patterns based on two metrics that have been implemented in the NMF R package, including Cophenetic and Silhouette[90]. Both metrics measure the stability for a particular number of patterns based on a hierarchical clustering of the consensus matrix. For a range of the number of patterns, a suitable number of patterns is the one at which Cophenetic and Silhouette values begin to drop suddenly.

In sum, the matrix $\mathbf{W}$ represents the $R$ latent patterns of cell groups, indicating how these cell groups coordinate to send signals; the matrix $\mathbf{H}$ represents the $R$ latent patterns of ligand-receptor pairs or signaling pathways, indicating how these ligand-receptor pairs or signaling pathways work together to send signals; the connection of $\mathbf{W}$ with $\mathbf{H}$ predicts the key signals sent from certain cell groups. Similarly, we summarized the communication probability array $\mathbf{P}$ along the first dimension to infer the key signals received by certain cell groups, as well as their latent patterns. Together, outgoing patterns reveal how the sender cells (i.e., cells as signal sources) coordinate with each other, as well as how they coordinate with certain signaling pathways to drive communication. Incoming patterns show how the target cells (i.e., cells as signal receivers) coordinate with each other, as well as how they coordinate with certain signaling pathways to respond to incoming signals.

To intuitively show the associations of latent patterns with cell groups and ligand-receptor pairs or signaling pathways, we used alluvial plots implemented in the ggalluvial package (https://cran.r-project.org/web/packages/ggalluvial/index.html). We first normalized each row of $\mathbf{W}$ and each column of $\mathbf{H}$ to be [0,1], and then set the elements in $\mathbf{W}$ and $\mathbf{H}$ to be zero if they are less than 0.5. Such thresholding allows to uncover the most enriched cell groups and signaling pathways associated with each inferred pattern, that is, each cell group or signaling pathway is associated with only one inferred pattern. These thresholded matrices $\mathbf{W}$ and $\mathbf{H}$ are used as inputs for creating alluvial plots. To directly relate cell groups with their enriched signaling pathways, we set the elements in $\mathbf{W}$ and $\mathbf{H}$ to be zero if they are less than $1/R$ where $R$ is the number of latent patterns. By using a less strict threshold, more enriched signaling pathways associated each cell group might be obtained. Using a contribution score of each cell group to each signaling pathway computed by multiplying $\mathbf{W}$ by $\mathbf{H}$, we constructed a dot plot in which the dot size is proportion to the contribution score to show association between cell group and their enriched signaling pathways.

### Quantification of similarity among intercellular communication networks.
Two different similarity measures were used to quantify the similarity among intercellular communication networks. A functional similarity $S$ was calculated based on the overlap of communications via the Jaccard similarity defined by:

$$S = \frac{E(G) \cap E(G')}{E(G) \cup E(G') - E(G) \cap E(G')}, \quad (5)$$

where $G$ and $G'$ are two signaling networks and $E(G)$ is the set of communications in signaling network $G$. High degree of functional similarity indicates major senders and receivers are similar, and it can be interpreted as the two signaling pathways or two ligand-receptor pairs exhibit similar and/or redundant roles.

A structural similarity was used to compare their signaling network structure, without considering the similarity of senders and receivers, using a previously developed measure for structural topological differences[91]. The dissimilarity measure between signaling networks $G$ and $G'$ with the number of cell groups being $N$ and $M$, respectively, is calculated by:

$$D(G, G') = w_1 \sqrt{\text{JSD}(u_G, u_{G'})/\log 2} + w_2 \left| \sqrt{\text{NND}(G)} - \sqrt{\text{NND}(G')} \right| \\ + \frac{w_3}{2} \left( \sqrt{\text{JSD}(P_{\alpha G}, P_{\alpha G'})/\log 2} + \sqrt{\text{JSD}(P_{\alpha G^c}, P_{\alpha G'^c})/\log 2} \right) \quad (6)$$

where $G^c$ indicates the complement of $G$, and JSD is the Jensen–Shannon divergence and $NND$ is defined as:

$$\text{NND}(G) = \frac{\text{JSD}(P_1, \dots, P_N)}{\log(d+1)} \quad (7)$$

with $\text{JSD}(P_1, \dots, P_N) = \frac{1}{N} \sum_{i,j}^{N} p_i(j) \log(\frac{p_i(j)}{u_j})$ and $u_j = (\sum_{i=1}^{N} p_i(j))/N$ being the Jensen–Shannon divergence and the average of the $N$ distributions, respectively. $P_i = \{p_i(j)\}$ is the distance distribution in each cell group $i$, where $p_i(j)$ is the fraction of cell groups connected to cell group $i$ at distance $j$. $d$ is the signaling network's diameter. $\text{JSD}(u_G, u_{G'})$ measures the difference between the signaling networks' averaged cell group-distance distributions, $u_G$ and $u_{G'}$, and $\text{JSD}(P_{\alpha G}, P_{\alpha G'})$ measures the difference between the $\alpha$-centrality values of the signaling networks. $w_1$, $w_2$, and $w_3$ are the weights of each term with $w_1 + w_2 + w_3 = 1$. Similar to a previous study[91], we selected $w_1 = 0.45$, $w_2 = 0.45$, and $w_3 = 0.1$. The structural similarity $S$ was computed by one minus dissimilarity measure $D$.

### Manifold and classification learning of intercellular communication networks.
The manifold learning of the inferred intercellular communication networks consists of three steps. First, we built a shared nearest neighbor (SNN) similarity network $Gs$ of all signaling pathways, which was constructed by calculating the $k$-nearest signaling pathways of each signaling pathway using the calculated functional or structural similarity matrix $S$ of intercellular communication networks. The fraction of shared nearest signaling pathways between a given signaling pathway and its neighbors was used as weights of the SNN network. The number of nearest neighbors $k$ was chosen as the square root of the total number of signaling pathways. Second, we smoothed the similarity matrix $S$ using $Gs \times S$. This smooth process provides a better representation of the similarity between signaling pathways to allow filtering of the weak similarity (potentially noise-induced) and enhancing the strong similarity[82]. Finally, we performed uniform manifold approximation and projection (UMAP)[92] on the smoothed similarity matrix. To better visualize the similarity of intercellular communication networks, we used the first two dimensions of the learned manifold, where each dot in this two-dimensional space represents an individual signaling pathway or ligand-receptor pair.

Moreover, to group the signaling pathways based on their similarity of intercellular communication networks in an unsupervised manner, we performed $k$-means clustering of the first two components of the learned manifold. The number of signaling groups was determined according to the eigenvalue spectrum by analyzing the Laplacian matrix derived from a consensus matrix[5,12]. First, we performed $k$-means clustering multiple times for different values of $k$ (e.g., 2–10). Second, we constructed a consensus matrix representing the probability of two signaling pathways being in the same group across multiple values of $k$. We then pruned the consensus matrix by setting the elements to be zero if they are less than 0.3 to ensure better robustness to noise. Third, we estimated the number of

signaling groups by computing the eigenvalues of the associated Laplacian matrix of the constructed consensus matrix. More generally, the number of signaling groups is usually determined by the first or second largest eigenvalue gap (i.e., the difference between consecutive eigenvalues) based on the spectral graph theory[93].

**Classification of cells into groups**. CellChat provides built-in functions to classify cells into groups. Briefly, a SNN graph of all cells is first constructed via the calculation of the $k$-nearest neighbors (20 by default) for each cell based on the low-dimensional representation space (e.g., via principle component analysis and diffusion map analysis) of the scRNA-seq data. The low-dimensional representation space can be either provided by user or computed by CellChat. Next, cells are clustered into groups by applying the Louvain community detection algorithm[94] to the constructed SNN graph. The number of cell groups is determined either by user-input resolution parameter in the Louvain algorithm or by an eigenvalue spectrum by analyzing the Laplacian matrix derived from multiple runs of Louvain algorithm with different resolution parameters.

**Single-cell RNA-seq datasets, data preprocessing, and analysis**. Mouse skin wound dataset. We used our recently published scRNA-seq dataset from mouse skin wounds[23]. This dataset included 21,819 cells and was generated via 10X Genomics platform (GEO accession code: GSE113854). Briefly, scRNA-seq was performed on unsorted cells isolated from mouse skin wound dermis from day 12 post-wounding. Unsupervised clustering identified fibroblasts (FIB, ~65%), immune cell populations, including myeloid cells (MYL, 15%), T lymphocytes (TCELL, 4%), B lymphocytes (BCELL, 3%), dendritic cells (DC, 1%), endothelial cells (ENDO, 9%), lymphatic endothelial cells (LYME, 1%), Schwann cells (SCH, 1%) and red blood cells (RBC, 1%). For the intercellular communication analysis, we excluded red blood cells and used the remaining 21,557 cells. The digital data matrices were normalized by a global method, in which the expression value of each gene was divided by the total expression in each cell and multiplied by a scale factor (10,000 by default). These values were then log-transformed with a pseudocount of 1. Normalized data were used for all the analyses. To investigate the heterogeneity of intercellular communications among different cell subpopulations, we performed subclustering analysis on the cell types, whose abundance in the dataset was greater than 5% using the Louvain community detection method. The number of cell groups was determined by the eigengap approach.

Embryonic mouse skin dataset. Recently published embryonic mouse skin datasets[22] were downloaded from GEO (accession codes: GSM3453535, GSM3453536, GSM3453537, and GSM3453538) and included two Embryonic day E13.5 biological replicates and two Embryonic day E14.5 biological replicates. These samples contain unsorted whole-skin cells captured via 10X Genomics platform. For both E13.5 and E14.5 scRNA-seq datasets, we removed the cells with the amount of UMI count less than 2500 and greater than 50000, as well as the cells with the number of genes less than 1000 and the fraction of mitochondrial counts greater than 20%. 12,951 cells at E13.5 and 12,197 cells at E14.5 were used for downstream analyses. First, we performed clustering analysis of cells from E13.5 and E14.5 using the Louvain community detection method, respectively. The values of the resolution parameter in the Louvain community detection method were explored to produce the major cell populations in embryonic skin[22,95]. Thus, 11 and 13 cell populations were identified at E13.5 and E14.5, respectively (Supplementary Fig. 3a–d). The cell populations were annotated based on the known markers[22,95]. Compared to E13.5, there were two specific populations at E14.5, including dermal condensate (DC) cells and pericytes. Second, we performed subclustering analysis of DC cells, basal cells and melanocytes at E14.5, respectively. This analysis identified three DC states including pre-DC, DC1, and DC2, three basal state including basal, proliferative basal and placode cells, and three melanocyte subpopulations including MELA-A, MELA-B, and MELA-C (Supplementary Fig. 3e-g). Third, we performed pseudotime analysis on epidermal and dermal cells at E14.5 using diffusion map, respectively.

Human disease skin dataset. The processed transcriptomic data of 17,349 cells from four lesional and four non-lesional human skin samples (patient ID: S1, S2, S3, S5, S7, S11, S14, and S15) from patients with atopic dermatitis was downloaded from GEO database under accession code GSE147424[70]. We performed the integration analysis of these eight samples using Seurat V3 package based on the tutorial from https://satijalab.org/seurat/v3.2/immune_alignment.html. Unsupervised clustering analysis segregated these combined cells into 10 broad cell types (Supplementary Fig. 8a-b), including fibroblasts (FIB), dendritic cells (DC), and T cells (TC). The original study highlighted the cell–cell communication among fibroblasts, dendritic cells and T cells[70]. Therefore, following the analysis from the original study[70], we performed the second-level clustering analysis of FIB, DC, and TC. FIB was clustered into five subgroups with distinct markers, including APOE high FIB (APOE + FIB), FBN1 + FIB, COL11A + FIB, Inflam.FIB (inflammatory FIB expressing chemokines such as CCL19) and a small T cell group expressing CD3D (Supplementary Fig. 8c). This contaminated TC population was removed for further analysis. DC was also separated into five subgroups, including cDC1 (type A DC), cDC2 (type B DC), LC (Langerhans cells), Inflam.DC (inflammatory DC) and other immune cells that does not express DC markers such as CD1A and CD1C (Supplementary Fig. 8d). This contaminated immune cell group was also removed for further analysis. TC was clustered into four subgroups, including TC, Inflam.TC (inflammatory TC), CD40LG + TC and NKT (NK

T cells) (Supplementary Fig. 8e). Together, CellChat was applied to 7563 cells from lesional and nonlesional skin involved in twelve cell groups, including APOE + FIB, FBN1 + FIB, COL11A + FIB, Inflam.FIB, cDC1, cDC2, LC, Inflam.DC, TC, Inflam.TC, CD40LG + TC, and NKT.

**Method comparisons**. We compare the performance of CellChat with three other tools, including SingleCellSignalR[9], iTALK[10], and CellPhoneDB[16] . We compare our database CellChatDB with other existing analogous databases, including CellTalkDB[71], CellPhoneDB[16], iTALK[10], SingleCellSignalR[9], Ramilowski2015[72], NicheNet[13] and ICELLNET[73]. SingleCellSignalR scores a given ligand-receptor interaction between two cell populations using a regularized product score approach based on average expression levels of a ligand and its receptor and an ad hoc approach for estimating an appropriate score threshold. iTALK identifies differentially expressed ligands and receptors among different cell populations and accounts for the matched ligand-receptor pairs as significant interactions. Cell-PhoneDB v2.0 predicts enriched signaling interactions between two cell populations by considering the minimum average expression of the members of the heteromeric complex and performing empirical shuffling to calculate which ligand–receptor pairs display significant cell-state specificity. The detailed description of how these methods were performed is available in Supplementary Note 3.

Both CellChat and CellPhoneDB, but not SingleCellSignalR, and iTALK, consider multi-subunit structure of ligands and receptors to represent heteromeric complexes accurately. To evaluate the effect of neglecting multi-subunit structure of ligands and receptors, we compute false positive rates for the tools that use only one ligand and one receptor gene pairs. The false positive interactions are defined by the interactions with multi-subunits that are partially identified by iTALK and SingleCellSignalR. The ground truth of the interactions with multi-subunits is based on our curated CellChatDB database. For example, for Tgfb1 ligand and its heteromeric receptor Tgfbr1/Tgfbr2 curated in CellChatDB, if the method only identifies one of the two pairs (Tgfb1–Tgfbr1 and Tgfb1–Tgfbr2), then we consider this prediction as one false positive interaction.

We performed subsampling of scRNA-seq datasets using a 'geometric sketching' approach, which maintains the transcriptomic heterogeneity within a dataset with a smaller subset of cells[96]. We evaluated the robustness of inferred interactions from subsampled datasets using three measures, including TPR, FPR, and ACC, which were defined in Supplementary Note 3. Note that such subsampling analysis was used to evaluate the consistency rather than accuracy.

**RNAscope in situ assay**. Frozen E14.5 mouse skin tissue sections were used for RNA in situ hybridization using RNAscope® kit v2 (323100, Advanced Cell Diagnostics). The following mouse probes from Advanced Cell Diagnostics were used: Dct probe (460461-C2), Edn3 (505841), Ednrb (473801-C3), Axl (450931-C2), Thy1 (430661-C3). We have complied with all relevant ethical regulations for animal testing and research. All animal experiments have been approved by the International Animal Care and Use Committee (IACUC) of the University of California, Irvine.

**Reporting summary**. Further information on research design is available in the Nature Research Reporting Summary linked to this article.

## Data availability

CellChatDB is included in the CellChat repository (https://github.com/sqjin/CellChat). KEGG pathway database is available at https://www.genome.jp/kegg/pathway.html. The datasets analyzed in this study are available from the Gene Expression Omnibus (GEO) repository under the following accession numbers: GSE113854, GSE122043 (including four samples GSM3453535, GSM3453536, GSM3453537, GSM3453538;) and GSE147424.

## Code availability

CellChat is publicly available as an R package. Source codes, as well as tutorials have been deposited at the GitHub repository (https://github.com/sqjin/CellChat). The web-based CellChat Explorer, including Ligand-Receptor Interaction Explorer for exploring the ligand-receptor interaction database and Cell–Cell Communication Atlas Explorer for exploring the intercellular communications in tissues, is available at http://www.cellchat.org/.

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

## Acknowledgements

This work was supported by the NSF grant DMS1763272, grant from the Simons Foundation (594598, QN), NIH grants U01AR073159, R01GM123731, and P30AR07504, Pew Charitable Trust (MVP), LEO Foundation grants LF-OC-20-000611 and LF-AW_RAM-19-400008 (MVP). C.F.G.-J. is supported by UC Irvine Chancellor's ADVANCE Postdoctoral Fellowship and a gift from the Howard Hughes Medical Institute Hanna H. Gray Postdoctoral Fellowship Program.

## Author contributions

S.J., M.V.P. and Q.N. conceived the project. M.V.P. and Q.N. supervised the research. S.J., M.V.P. and C.F.G.-J. curated the database. S.J. and L.Z. developed and implemented the computational approach. S.J., C.F.G.-J., L. Z, M.V.P. and Q.N. performed data analysis. R.R. and C.-H.K. performed and analyzed RNAscope experiments. I.C., M.V.P. and S.J. developed the web interface. S.J. and C.F.G.-J. prepared the figures. S.J., M.V.P. and Q.N. wrote the manuscript. P.M. edited the manuscript. All authors read and approved the final manuscript.

## Competing interests

The authors declare no competing interests.
