## [Peer Review File · Nature Communications]

REVIEWER COMMENTS

Reviewer #1 (Remarks to the Author):

In this paper, the author developed an open source R package to infer, visualize and analyze the intercellular communications from scRNA-seq data called CellChat based on a manually curated comprehensive signaling molecule interaction database. CellChat can not only infer cell-state specific signaling communications within a given scRNA-seq data using mass action models, but also provides several visualization outputs to facilitate intuitive user-guided data interpretation. However, I have several concerns on the publication of the paper:

1. As CellChat is heavily based on the underlying database of ligand-receptor pairs, the data quality of the database is very important. The description of database construction is too simple, the author should describe more details of database construction including how to collect ligand-receptor pairs from KEGG step by step, how to define coligands or coreceptors and their interacting partners, and the manual collection from other resources or literatures. If possible, the related code should be provided for reproduction of CellChatDB.
2. As the limitations of current existing databases, the author should list the statistical data of CellChatDB and other known LR databases, such as SingleCellSignalR, CellPhoneDB, NicheNet, iTALK, etc. to summarize the difference and strength of CellChatDB. Similarly, the author should also compare systematically the inferred intercellular communication network with other methods (SingleCellSignalR, CellPhoneDB, NicheNet, iTALK, etc.) on a specific dataset, e.g., the mouse skin scRNA-seq dataset, including the inferred significant ligand-receptor pairs and inferred cell-cell communications.
3. As CellChatDB includes curated ligand-receptor pairs of human and mouse, all cases in the manuscript are mouse scRNA-seq datasets. The author should also test CellChat on a human scRNA-seq dataset by using human ligand-receptor pairs.
4. For ligand-receptor pair, it is easy to understand that when the ligand is highly expressed in one cell group and the receptor is highly expressed in another cell group, the interaction between these two cell groups may occur and can be inferred. However, ligand-coreceptors pairs and coligand-receptor pairs are more complex, how to define the interaction by three or more genes involving ligand-coreceptors pairs and coligand-receptor pairs? What's more, difference and detail of how to treat ligand-receptor pairs, ligand-coreceptors pairs, and coligands-receptor pairs when calculating intercellular communication probability should be stated clearly in Methods.
5. In Figure 2A, the hierarchical plot shows the inferred intercellular communication network for TGFb signaling. However, this plot only shows the FB-related communications with other cell groups, while other intercellular communications cannot be obtained, for example, the communications between DC-LYME, MYL-ENDO, etc. It is necessary and informative if CellChat can show the overview of inferred intercellular communication network comprehensively for each pathway.
6. In Figure 2A, the author should explain the meaning of the edge size in the Figure legend.

Reviewer #2 (Remarks to the Author):

This paper by Nie and colleagues contains two main contributions. First, they created a database, called CellChatDB for multimeric ligand-receptor complexes. Second, they developed a new computational method for inferring active ligand-receptor pathways from single-cell RNAseq data. Both the database and the method could be valuable resources for the community.

Major concerns:

- The main source of information underlying CellChatDB already exists in the widely used database KEGG. It is unclear what 'manual curation' means in this context or what additional value CellChatDB provides other than a visualization interface.
- The computational method proposed here lack sufficient validation and benchmarking. Predictions are made from single-cell RNAseq data from mouse skin cells, where the ground-truth is unknown. To circumvent this difficulty, the authors used a subsampling approach, treating the results obtained from the full dataset as the ground-truth against which the predictions from subsamples are compared. This is a flawed assumption. In reality, this approach only evaluates sensitivity to input data, whereas errors intrinsic to model assumptions would be inherited across all datasets therefore cannot be detected in this way.
- In evaluation of different methods, the authors assume that agreement between multiple methods is proxy for accuracy. This assumption is also flawed, because similar methods tend to generate similar results regardless of accuracy. The bottom line is that, without external curated information as a guide, it is impossible to evaluate the performance of different methods. Presumably the CellChatDB database can be used here to aid model evaluation, but it is unclear why they didn't proceed in this direction.
- The authors made a number of interesting predictions regarding the cell-type specific signaling pathways in mouse skin in response to wound healing and during embryonic development, but these predictions could be more substantiated if followed by experimental validation.

Minor concerns:

- Cell clustering is a pre-requisite for cell-cell interaction prediction, but clustering results can be different depending on which clustering methods are used and which parameters are chosen. How does such uncertainty affect the outcome of ligand-receptor interactions?
- Related to the previous comment, nine fibroblast cell types were identified in mouse skin wound tissue. Are they truly distinct cell types? FIB-D seems to have unique signaling properties than the others. Have such specialized fibroblast cell types been observed before?
- The analysis of incoming and outgoing signaling patterns seems interesting, but it is unclear what is the distinction between incoming and outgoing patterns. Is there a mechanistic interpretation for these patterns?
- False positives rate is used to evaluate the performance of italk and singlecellsignalR. How is this calculated exactly? The first sentence on page 27 is vague and seems to be associated with consistency rather than accuracy. What is the ground truth? What is the false positive rate of the method presented in this study? Why is CellPhoneDB not compared in this analysis?
- On a practical side, the cellchat website posts predicted results in mouse skin, but a user might be interested to apply this method to analyze their own data. Is this possible?

Reviewer #3 (Remarks to the Author):

In the manuscript Inference and analysis of cell-cell communication using CellChat, Jin et al. presented a database of interactions among ligand and receptors, and a methodology to infer inter-cellular communication networks. There are a few subsequent analyses based on the inferred network, including centrality related concepts like dominant senders/receivers/mediators, etc., as well as using non-negative matrix decomposition to break down the network into distinct patterns. The manuscript has provided the community an alternative way and great visualization tools for inferring and comparing cell-cell communication between different biological conditions using scRNA-seq data. However, as a methodology paper, some data provided in the manuscript should be further validated to support their conclusions and need to be carefully addressed.

Major points:

1. Lies at the core is an ambitious model law-of-mass-action model that takes into account almost everything: the ligand/receptor expression and what they form multi-units complexes, co-stimulatory, and co-inhibitory ligands/receptors, roles of agonist and antagonist. While all these components and their interactions are relevant, modeling of all these mechanistic processes requires a high level of details which is very hard to achieve with scRNA-seq data. Remember LMA happens in protein level, not in RNA level. So ironically, the model seems to capture everything

but there are many assumptions, almost all parameters are arbitrarily chosen, and not easy to justify. For instance, why the dissociation constant is always 0.5? Why the Hill coefficient is always 1? Such a detailed model will make the study a lot more depends on the correctness of the curated database.

2. There's no doubt the authors did their best to construct CellChatDB, but there's no perfect source of information. I think, at the very minimal, the authors should do the following test to show that the outputs of their model capture a certain level of real signals rather than purely noise: In several ways randomize their curated database, like ligand-receptor interactions, the corresponding agonist/antagonist, co-receptors, etc., one-by-one and in some combinations, and then repeat the identification of statistically significant communications. If the number of significant pairs identified using the fake database is similar, then bad news, suggesting the outputs are simply false positive. In fact, the procedure could provide a way to quantify the false discovery rate.

3. In revealing continuous cell lineage-associated signaling events, the authors predict the cell-cell communication in the different stages during pseudotime and real embryonic stages (E13.5/E14.5). However, the authors should provide expression pattern of ligands and receptors in all the predicted interactions during skin cell development side by side to validate their prediction results. Based on methods, we should be able to see a similar pattern of expression of ligand/receptors with communication probability of predicated interactions during developmental stages.

4. In predicting key signaling events between spatially colocalized cell populations, the authors used spatially-colocalized 4 cell types to showcase their prediction on the cell-cell communication. However, proper controls are not provided to validate the predictions. The authors should add in cell types that are not spatially adjacent to these 4 cell types to the same analysis and demonstrate that cell-cell communication identified in Figure 4 are stronger in spatially adjacent cell types but not in spatially distant cell types.

5. In comparison with other cell-cell communication inference tools, current metrics used to compare the tools by reasoning that a more accurate method will have a larger proportion of overlapped predictions with other methods is not reasonable and the result is not convincing. The ligand receptor databases used in CellChat and CellPhoneDB are different which will directly contribute to the number and variety of predicted interactions. Besides, the cell interactions identified by SingleCellSignalR and iTalk but not CellChat, due to failed detection of interactions with multi-subunits, are not necessarily 'false-positive', which could be caused by low expression of multi-subunits of the receptors. The authors should use better metrics to compare those inference tools, for example, whether these tools can correctly capture stronger interaction in spatially adjacent cells but not spatially distant cells. In Supplementary Figure 8(a), the authors overlapped ligand-receptor interactions between CellChat/CellPhoneDB and other two methods including SingleCellSignalR and iTALK. CellChat should be compared with CellPhoneDB in terms of overlapping interaction. In Supplementary Figure 8(b), CellChat is not outperforming CellPhoneDB much even with the modeling of almost everything. The authors need to explain this in "Method comparisons" section.

6. The authors adopted non-negative matrix factorization for the identification of major signals of specific cell groups and global communication patterns. The number of patterns 5 seems to be random or experiential. Without knowing pattern's biological meaning, it is unrealistic to guess the real number of patterns even with the domain knowledge. In page 10, the authors claimed that they can predict the sequential signaling events of cells, e.g., FIB-A cell secreted EGF and GALECTIN signals first. Then FIB-D and FIB-E coordinate... It is easier to identify groups of cell types and signals, but how this time-series event is inferred from the patterns is unclear.

A few minor points:

1. While P_{ij}^k is likely to lie between 0 to 1 (because of the last term), it is not exactly a probability, in the sense, $\sum_j P_{ij}^k$ might not be 1. Should there require certain normalization? Or simply say the quantity scales with the probability?

2. Because of dropout, 0 is quite common in scRNA-seq data. I am slightly worried about estimating the level of ligands by the geometric mean of the sub-units. Similarly, for the robust measure of average gene expression based on Q1, Q2, Q4. So, if there are more than 25% of dropout, EM=0. Can the authors provide some statistics on how many pairs are dropped?

3. About the non-negative matrix factorization step, the authors reduce the 3D array P to 2D by summing over the receivers so that NMF could be used. It erases patterns associated with the receiver-end. What happens if we sum over the sources? Have the authors considered tensor decomposition?

4. The standard of good figure legends is that the readers can easily get an idea of the figures without going back and forth among main text, figures and methods. The authors should improve their figure legends and clearly demonstrate what they did in each figure, instead of just generally saying what kind of plot/diagram it is.

REVIEWER COMMENTS

Reviewer #1

In this paper, the author developed an open source R package to infer, visualize and analyze the intercellular communications from scRNA-seq data called CellChat based on a manually curated comprehensive signaling molecule interaction database. CellChat can not only infers cell-state specific signaling communications within a given scRNA-seq data using mass action models, but also provides several visualization outputs to facilitate intuitive user-guided data interpretation. However, I have several concerns on the publication of the paper:

Response: We thank the reviewer for the insightful comments. Our detailed responses are provided below. Substantial improvement has been made in the revision, and multiple changes highlighted with red were introduced throughout the manuscript.

R1-1. *As CellChat is heavily based on the underlying database of ligand-receptor pairs, the data quality of the database is very important. The description of database construction is too simple, the author should describe more details of database construction including how to collect ligand-receptor pairs from KEGG step by step, how to define coligands or coreceptors and their interacting partners, and the manual collection from other resources or literatures. If possible, the related code should be provided for reproduction of CellChatDB.*

Response: This is an important point. In the revised manuscript, we have significantly expanded the description of the database construction and also provided instructions on how to update CellChatDB using user-defined ligand-receptor interactions (see revised Supplementary Text). Since CellChatDB is a manually curated database of ligand-receptor interactions constructed by carefully reviewing the KEGG pathway maps and recent peer-reviewed studies, we do not have codes for automatically creating database. However, we provide codes for updating CellChatDB using user-defined lists of curated signaling interactions.

We apologize for the confusing terminologies used in the Methods section. The co-factors we considered include: soluble agonist, soluble antagonist, co-stimulatory and co-inhibitory membrane-bound receptors. The co-ligands are either soluble agonist or antagonist and, thus, we removed the terminology of co-stimulatory ligands and co-inhibitory ligands. Since soluble agonist, antagonist and co-receptor usually either enhance or attenuate the main ligand-receptor interaction to modulate signaling, we did not consider ligand-co-receptors pairs and co-ligands-receptor pairs. We clarified the terminologies in the revised Methods section (Page 27).

The revised manuscript provides more details on how CellChatDB was built in Supplementary Text. CellChatDB is a database of literature-supported ligand-receptor interactions in mouse and human. The majority of ligand-receptor interactions were manually curated on the basis of KEGG signaling pathway database, and additional signaling molecular interactions were gathered from recent peer-reviewed studies. In particular, the detailed steps for collecting ligand–receptor interactions in mouse are as follows.

Step 1: We collected the list of all major signaling pathway families, which are related to “Signal Transduction” and “Signaling Molecules and Interaction” (<https://www.genome.jp/kegg/pathway.html>) in KEGG pathway database.

Step 2: We manually curated ligand-receptor interactions by reviewing all relevant KEGG pathway maps. For each map, e.g., TGF β signaling pathway (Supplementary Figure S11a), each signaling molecule was classified as one of six categories: ligand, receptor, agonist, antagonist, co-stimulatory and co-inhibitory membrane-bound receptors. Signaling role of each molecule is clearly indicated in KEGG pathway maps. Note that agonist, antagonist, co-stimulatory and co-inhibitory receptors are considered as cofactor molecules, which only modulate ligand-receptor mediated signaling strength, and do not produce new ligand-receptor pairs. After a ligand-receptor pair was chosen, we determined its co-factor molecules. For example, for Tgfb1 ligand (green box) and its receptor complex Tgfb1/Tgfb2 (pink box) (new Supplementary Figure S11a), Thbs1

(light orange box) is found to be an agonist because Thbs1 inhibits Ltbp1 and Lrbp1 inhibits the ligand Tgfb1, implying a positive regulation of Tgfb1 by Thbs1. For this case, antagonists include Ltbp1, Lefty1, Fmod, Dcn (orange box) because they are negative regulators of Tgfb1. Co-inhibitory receptor is Bambi (purple) because of its negative role as a membrane-bound receptor. Similarly, we collected other ligand-receptor pairs as well as cofactors by reviewing all available KEGG pathway maps. Supplementary Figure S11a shows several other examples of ligand-receptor pairs and how we defined ligands, (multi-subunit) receptors and cofactors. Note that we did not consider ligand-co-receptors pairs, agonist-receptor pairs and antagonist-receptor pairs because these co-factors usually either enhance or attenuate the main ligand-receptor interaction to modulate signaling.

Step 3: We carefully reviewed several signaling molecule/compound families in the KEGG database, including Cytokine Receptors (ko04050), Cytokines and Growth Factors (ko04052), and Bioactive Peptides (br08005), to ensure that these ligands/receptors are included in the curated ligand-receptor pairs in Step 2.

Step 4: We collected additional ligand-receptor interactions by reviewing the known signaling pathway families in recent peer-reviewed studies. For example, for TGF β signaling pathway, we collected nine additional ligand-receptor pairs from two prominent papers (PMIDs: 29376829, 27449815). A recent experimental study showed that TGF- β uses a novel mode of receptor activation (i.e., TGFBR1 and ACVR1) to phosphorylate SMAD1/5 (PMID: 29376829; *Elife* 2018). Another study also showed that TGF- β ligands can bind to type I receptor ACVR1B/ACVR1C (PMID: 27449815; *Cold Spring Harbor Perspectives in Biology* 2016). The evidence for each literature-supported ligand-receptor interaction is provided in CellChatDB. Together, the mouse CellChatDB contains 2,021 validated molecular interactions, including 60% of paracrine/autocrine signaling interactions, 21% of extracellular matrix (ECM)-receptor interactions and 19% of cell-cell contact interactions.

To collect signaling interactions in human, mouse gene symbols in CellChatDB were mapped to human orthologues using the human orthologue information via MGI (<http://www.informatics.jax.org/homology.shtml>). We also manually added signaling interactions specific to human, including CXCL8, CCL13, CCL14, CCL15, CCL16, CCL23, XCL2, IFNW1, IL1F7, IL26, IL29. In sum, human CellChatDB contains 1,939 validated molecular interactions, including 61.8% of paracrine/autocrine signaling interactions, 21.7% of extracellular matrix (ECM)-receptor interactions and 16.5% of cell-cell contact interactions.

R1-2. *As the limitations of current existing databases, the author should list the statistical data of CellChatDB and other known LR databases, such as SingleCellSignalR, CellPhoneDB, NicheNet, iTALK, etc. to summarize the difference and strength of CellChatDB. Similarly, the author should also compare systematically the inferred intercellular communication network with other methods (SingleCellSignalR, CellPhoneDB, NicheNet, iTALK, etc.) on a specific dataset, e.g., the mouse skin scRNA-seq dataset, including the inferred significant ligand-receptor pairs and inferred cell-cell communications.*

Response: Thank you for the suggestion. In the revision we compared CellChatDB with six other known ligand-receptor (L-R) databases:

- CellPhoneDB (Efremova et al., *Nat Protoc* 2020),
- iTALK (Wang et al., *bioRxiv* 2019),
- SingleCellSignalR (Cabello-Aguilar et al *Nucleic Acids Res* 2020),
- Ramilowski2015 (Ramilowski et al. *Nat Commun* 2015),
- NicheNet (Browaeys et al., *Nat Methods* 2019),
- ICELLNET (Noël et al., *bioRxiv* 2020).

Among these databases, only CellChatDB contains ligand-receptor interactions in mouse. Therefore, we performed comparison of the human version of CellChatDB with the above databases by counting the number of L-R pairs, L-R pairs with multi-subunits, and L-R pairs with co-factors (e.g., soluble agonist, antagonist, co-stimulatory and co-inhibitory receptors) (new Supplementary Figure 11b). First, the number of L-R pairs in CellChatDB

(i.e., 1,939 pairs) is higher than both in CellPhoneDB and ICELLNET, but lower than in other four databases, among which NicheNet has the largest number of L-R pairs (i.e., 12,651 pairs). Second, the number of L-R pairs with multi-subunits in CellChatDB (i.e., 928 pairs) is twice larger than in CellPhoneDB and three times larger than in ICELLNET, whereas other four databases do not consider L-R pairs with multi-subunits. Third, only CellChatDB contains L-R pairs with co-factors. Taking into an account subunit structure of ligands and receptors is essential because cell-cell communication often relies on multi-subunit protein complexes. Cofactors may also modulate cell-cell communication both positively and negatively. Therefore, CellChatDB provides an important resource for identifying biologically meaningful cell-cell communication. In the revision we have added these points in the Results (Page 21) and Supplementary Text (Page 4).

In addition, we have systematically compared the inferred cell-cell communication networks by using three other methods (CellPhoneDB, iTALK, SingleCellSignalR) using an example of four spatially colocalized cell populations in embryonic mouse skin (revised Figure 4). We excluded NicheNet, because it does not give a clear-cut answer about whether ligand-receptor pairs are significant, as mentioned in its tutorial on Github (<https://github.com/saeyslab/nichenetr/blob/master/vignettes/faq.md>). We compared the inferred significant ligand-receptor (L-R) pairs for any two cell subpopulations between CellChat and other methods. Specifically, we calculated the number of L-R pairs inferred by each method, and the number of shared L-R pairs between any two methods (Supplementary Table S2). The average numbers of L-R pairs inferred by CellChat, CellPhoneDB, iTALK and SingleCellSignalR between two cell subpopulations were 12, 37, 14 and 12, respectively. Note that we retained the top 10% of L-R pairs (most significant pairs) inferred by iTALK and SingleCellSignalR. We found that CellChat shared more L-R pairs with CellPhoneDB than with iTALK, likely due to the fact that both CellChat and CellPhoneDB consider the multi-subunit complexes and determine the significant L-R pairs using a statistical approach. SingleCellSignalR shared very few L-R pairs with other three methods, suggesting a potentially different logic for quantifying and ranking L-R interactions. One major finding is that the majority of shared L-R pairs between CellChat and CellPhoneDB were independently ranked as top pairs by CellPhoneDB

(Supplementary Data 1). This result suggests that although CellChat infers fewer L-R pairs than CellPhoneDB, it captures the strongest (and likely the most significant) L-R interactions. The inferred L-R pairs for each method between any two subpopulations are listed in the new Supplementary Data 1. In the revised manuscript we have added these new results in the Results (Page 19)

R1-3. *As CellChatDB includes curated ligand-receptor pairs of human and mouse, all cases in the manuscript are mouse scRNA-seq datasets. The author should also test CellChat on a human scRNA-seq dataset by using human ligand-receptor pairs.*

Response: Thank you for the suggestion. In the revision, we added CellChat analysis on human dataset. Specifically, we studied signaling changes between lesional (diseased) and nonlesional (normal) skin from patients with atopic dermatitis (AD) using recently published human skin scRNA-seq dataset (PMID: 32035984; Helen et al., *J Allergy Clin Immunol* 2020). The original study revealed that lesional skin was enriched for chemokine signals (including CCL19) from inflammatory fibroblasts to inflammatory immune cells, including dendritic cells (DC) and T cells (TC). This was validated using immunofluorescence staining. Therefore, we used CellChat to study the intracellular communication among fibroblasts (four subpopulations: APOE+ FIB, FBN1+ FIB, COL11A+ FIB and Inflam.FIB), DCs (four subpopulations: cDC1, cDC2, LC and Inflam.DC), and TCs (four subpopulations: TC, Inflam.TC, CD40LG+ TC and NKT) (new Supplementary Fig. 12). By comparing the overall communication probability between nonlesional and lesional skin, we found that 11 out of 16 signaling pathways were highly active in lesional skin, including 9 pathways involved in inflammatory and immune response, such as CXCL, LIGHT, GLAECTIN, COMPLEMENT, MIF, CSF, IL4, CCL and TNF (new Figure 7c). Four pathways were specifically active in lesional skin, including known inflammatory signals CSF, IL4, CCL and TNF. Specific to CCL signaling, CellChat identified ligand-receptor pair CCL19-CCR7 as the most significant signaling, contributing to the communication from Inflam.FIB to Inflam.DC (new Figure 7d-f). This is in agreement with the exported experimental finding (Helen et al., *J Allergy Clin Immunol* 2020). Ligand MIF and its multi-subunit receptor CD74/CD44 were found to act as major

signaling from Inflammation.FIB to Inflammation.TC in lesional skin compared to nonlesional skin (new Figure 7d and Supplementary Fig. 13). Together, CellChat's joint analysis using an example of human lesional and nonlesional skin enables the discovery of major signaling changes that might drive disease pathogenesis. We have added these results in a new subsection of the revised Results section (Page 17).

R1-4. *For ligand-receptor pair, it is easy to understand that when the ligand is highly expressed in one cell group and the receptor is highly expressed in another cell group, the interaction between these two cell groups may occur and can be inferred. However, ligand-coreceptors pairs and coligand-receptor pairs are more complex, how to define the interaction by three or more genes involving ligand-coreceptors pairs and coligand-receptor pairs? What's more, difference and detail of how to treat ligand-receptor pairs, ligand-coreceptors pairs, and coligands-receptor pairs when calculating intercellular communication probability should be stated clearly in Methods.*

Response: We apologize for the confusion. As mentioned above, co-ligands are either soluble agonists or antagonists and, thus, we updated our terminology in the revised manuscript. Since co-ligand and co-receptor either enhance or attenuate the main ligand-receptor interaction to modulate signaling, we did not consider ligand-co-receptors pairs and co-ligands-receptor pairs. In contrast, we modeled how these soluble agonists, antagonists and co-receptors increase or decrease the intercellular communication probability. For each ligand-receptor pair with multiple soluble agonists, we computed the average expression of these agonists (denoted by AG) and then used a well-studied Hill function, which is widely used to describe positive regulations in signal transduction, to model the positive modulation of the ligand-receptor interaction. For each ligand-receptor pair with multiple soluble antagonists, we modeled them using the same approach. For each ligand-receptor pair with multiple co-stimulatory receptors, we computed the average expression of these co-stimulatory receptors (denoted by RA) and then used a linear function to model the positive modulation of the receptor expression. We clarified these details in the revised Methods (Page 27).

R1-5. In Figure 2A, the hierarchical plot shows the inferred intercellular communication network for TGFb signaling. However, this plot only shows the FB-related communications with other cell groups, while other intercellular communications cannot be obtained, for example, the communications between DC-LYME, MYL-ENDO, etc. It is necessary and informative if CellChat can show the overview of inferred intercellular communication network comprehensively for each pathway.

Response: Sorry for the confusion. The hierarchical plot intends to provide an overview of inferred intercellular communication network for each pathway, consisting of two major parts: the left portion shows signaling to FIBs coming from either FIBs or immune/blood vessel cells (e.g., MYL, DC, ENDO); and the right portion shows signaling to immune/blood vessel cells coming from either FIBs or immune/ blood vessel cells. Therefore, the right portion shows communications between immune/blood vessel cells, such as DC-LYME and MYL-ENDO. In the revised manuscript, we have clarified this in the Results and Figure legends to emphasize the difference between left and right parts. In addition to the hierarchical plot, CellChat provides a circle plot for visualizing the global intercellular communication network. For most cases, the customized hierarchical plot tends to provide a more comprehensible way to visualize oftentimes complex details of signaling by a given pathway.

R1-6. In Figure 2A, the author should explain the meaning of the edge size in the Figure legend.

Response: Sorry for the confusion. The edge size is proportional to the interaction strength, i.e., the inferred communication probability. In the revised manuscript, we provided this information in the Figure legends.

Reviewer #2

This paper by Nie and colleagues contains two main contributions. First, they created a database, called CellChatDB for multimeric ligand-receptor complexes. Second, they developed a new computational method for inferring active ligand-receptor pathways from single-cell RNAseq data. Both the database and the method could be valuable resources for the community.

Response: We thank the reviewer for appreciating the importance of CellChat method and for the insightful comments. Below please find our detailed responses. Substantial improvement has been made in the revision, and multiple changes highlighted with red were introduced throughout the manuscript.

R2-1. *The main source of information underlying CellChatDB already exists in the widely used database KEGG. It is unclear what ‘manual curation’ means in this context or what additional value CellChatDB provides other than a visualization interface.*

Response: This is an important point. Although KEGG database contains the ligand-receptor interaction information, such information is not “ready-to-use” when developing a systematical approach for inferring cell-cell communication. By incorporating carefully reviewed KEGG pathway maps and recent peer-reviewed studies, CellChatDB provides the following added value.

- It collects ligand-receptor pairs from KEGG signaling pathway maps.
- It categories signaling molecules in each KEGG pathway map into different groups based on their roles in ligand-receptor interaction: agonist, antagonist, co-stimulatory receptor and co-inhibitory receptor.
- It arranges all collected information into a structured data format, allowing for easy computational analysis and provides new community resource to develop a systematical approach for cell-cell communication analysis.
- It adds ligand-receptor interactions that are not in KEGG database through carefully reviewing known signaling pathways in recent peer-reviewed studies.

In the revised manuscript, we added the above points in the Supplementary Text.

In this revision we have also compared CellChatDB with other six known ligand-receptor (L-R) databases:

- CellPhoneDB (Efremova et al., *Nat Protoc* 2020),
- iTALK (Wang et al., *bioRxiv* 2019),
- SingleCellSignalR (Cabello-Aguilar et al. *Nucleic Acids Res* 2020),
- Ramilowski2015 (Ramilowski et al. *Nat Commun* 2015),
- NicheNet (Browaeys et al., *Nat Methods* 2019),
- ICELLNET (Noël et al., *bioRxiv* 2020).

Among these databases, only CellChatDB contains ligand-receptor interactions in mouse. Therefore, we compared the human version of CellChatDB with the above databases by counting the numbers of L-R pairs, L-R pairs with multi-subunits, and L-R pairs with co-factors (e.g., soluble agonist, antagonist, co-stimulatory and co-inhibitory receptors) (new Supplementary Fig. 11b). In general, CellChatDB provides an important resource for the community with added value to better develop biologically meaningful understanding of cell-cell communication. In the revision we have added these results in the Results section (Page 21) and Supplementary Text (Page 4).

Moreover, in this revision we have significantly expanded on the description of database construction and provided step-by-step details on how the ligand-receptor interactions were collected (see revised Supplementary Text).

R2-2. *The computational method proposed here lack sufficient validation and benchmarking. Predictions are made from single-cell RNAseq data from mouse skin cells, where the ground-truth is unknown. To circumvent this difficulty, the authors used a subsampling approach, treating the results obtained from the full dataset as the ground-truth against which the predictions from subsamples are compared. This is a flawed assumption. In reality, this approach only evaluates sensitivity to input data, whereas errors intrinsic to model assumptions would be inherited across all datasets therefore cannot be detected in this way.*

Response: We agree that subsampling approach only evaluates sensitivity to the input data. In our original manuscript, we actually did not use this subsampling approach to assess the accuracy of the predictions, but to assess the robustness of our method. In the revised manuscript, we have made this point clearer.

Due to the lack of ground-truth of the intracellular communication network, it remains challenging to systematically evaluate the predictions from any given computational methods. Similar to other existing methods, we validated the predictions based on the literature. From our experience working on skin morphogenesis and regeneration, we found that CellChat's predictions can recapitulate known biology to a substantial degree. In the revision we performed a systematic evaluation of different computational methods based on the assumption that spatially adjacent cell types should have stronger cell-cell communication than spatially distant cells. In the original manuscript we studied cell-cell communication for four spatially colocalized cell populations in embryonic mouse skin, including Placodes, pre-DC, DC1 and DC2 (Figure 4). We have now added in cell types that are likely not spatially adjacent to the above four cell types and updated our analysis. We then tested whether different computational tools can correctly capture stronger interactions in spatially adjacent cells.

We added seven cell types from embryonic day E14.5 mouse skin dataset, including FIB (fibroblasts), MELA (melanocytes), Spinous (spinous epithelial cells), MYL (myeloid cells), Immune (other immune cells), ENDO (endothelial cells) and Muscle. We then computed the number of inferred communications as well as the sum of communication probabilities between each cell type and the four spatially colocalized cell populations for each method. We found that CellChat consistently captures stronger interactions in spatially adjacent cells than spatially distant cells both in terms of the number of interactions and the interaction probabilities (new Supplementary Fig. 9a-b). CellPhoneDB also performed well at discriminating spatially adjacent cells from spatially distant cells. iTALK failed to capture stronger interactions in spatially adjacent cells as compared to spatially distant cells (for FIB, MELA, MYL and ENDO). SingleCellSignalR failed to capture stronger

interactions between spatially adjacent cells vs. spatially distant cells (for FIB and ENDO). In addition, by considering all seven cell types together, we found that both CellChat and CellPhoneDB can significantly distinguish the spatially adjacent cells from spatially distant cells, whereas iTALK and SingleCellSignalR failed to do so (new Supplementary Fig. 9c). Since CellPhoneDB infers more interactions than CellChat, we tested whether the top interactions predicted by CellPhoneDB can also distinguish the spatially adjacent from spatially distant cells. For the top 10%, top 20% and top 30% interactions predicted by CellPhoneDB, the difference between spatially adjacent and spatially distant cells was not as significant as with CellChat (new Supplementary Fig. 10a-b), suggesting that CellChat performed better at capturing stronger interactions. Together, although CellChat produces fewer interactions, it performs well at predicting stronger interactions. These results were added in revised Results section (Page 20).

R2-3. *In evaluation of different methods, the authors assume that agreement between multiple methods is proxy for accuracy. This assumption is also flawed, because similar methods tend to generate similar results regardless of accuracy. The bottom line is that, without external curated information as a guide, it is impossible to evaluate the performance of different methods. Presumably the CellChatDB database can be used here to aid model evaluation, but it is unclear why they didn't proceed in this direction.*

Response: Thank you for the insightful comment. We agree that agreement between multiple methods is not sufficient to determine accuracy. We have made this point clearer in the revised manuscript. We also agree that it is impossible to evaluate the performance of different methods without external curated information as a guide. However, we don't think CellChatDB database is very helpful for model evaluation because of the following reasons. CellChatDB and other existing databases contain only ligand-receptor pair information and lack cell type information that can be used to assess the inferred intracellular communications between two cell types. On the other hand, testing whether these methods can correctly capture stronger interactions in spatially adjacent cells but not in spatially distant cells could be a good way for evaluating different methods. In the

revision we have added a discussion on the evaluation and benchmarking of methods for inferring cell-cell communication (Page 24).

R2-4. *The authors made a number of interesting predictions regarding the cell-type specific signaling pathways in mouse skin in response to wound healing and during embryonic development, but these predictions could be more substantiated if followed by experimental validation.*

Response: In this revision we used multiplexed RNA *in situ* detection method (RNAscope) to validated two novel CellChat predictions (that were not previously described in the literature) on the signaling interactions during early hair follicle morphogenesis in developing embryonic skin. First prediction: CellChat revealed that at E14.5, DC cells respond to autocrine PROS pathway (Fig. 4g). Pros1 is the ligand for the pathway, that signals via receptor tyrosine kinase Axl. Signaling via Axl has been implicated in conferring cells with migratory properties in different biological context, including cancer invasion^{44, 45} and directional migration has been recently shown to be crucial for normal dermal condensate formation upon hair follicle morphogenesis⁴⁰. We examined CellChat's prediction of active PROS signaling in DC cells by RNAscope for *Edn3* as DC marker, *Axl* and *Thy1* (Cd90) as a marker of cell migration^{46, 47}. As expected from scRNA-seq, *Axl* was expressed broadly, including in *Edn3*+ DC, overlaying placode and surrounding epithelium. However, *Thy1* expression was concentrated in DC with significantly lower levels elsewhere (Fig. 4h). This RNAscope result is consistent with the possibility of autocrine PROS signaling in DC, likely driven via Pros1-Axl signaling.

Second prediction: During early hair follicle formation at E14.5, melanoblasts (melanocyte precursor cells) migrate into the hair placode from the dermis and then become differentiated toward melanocytes. However, the mechanisms of melanocyte migration into placode remain incompletely understood⁴⁸. Therefore, we further studied the cell-cell communication among placodes, DC cells and melanocyte cells (including three melanocyte subpopulations: MELA-A, -B and -C; see Methods). CellChat revealed that melanocytes strongly respond to DC cells via previously unrecognized EDN signaling (Fig. 4i). *Edn3* is the ligand for EDN pathway, that regulates melanocyte migration⁴⁹.

Therefore, CellChat prediction suggests DC cells induce early directed migration of melanocytes. To experimentally examine this prediction, we used RNAscope technique to spatially map expression of *Dct*, that marks late-stage melanocyte precursors, *Edn3* ligand and its receptor *Ednrb* in E14.5 embryonic mouse skin. As expected, *Dct*⁺ melanocytes (i.e., MELA-C subpopulation) localize inside the placode. They also express *Ednrb*. In turn, *Edn3* is specifically enriched in DC cells (preDC, DC1 and DC2 subpopulations), while *Ednrb* is also enriched in a portion of DC cells (likely DC2 subpopulation). Scattered *Ednrb*⁺/*Edn3*^{neg}/*Dct*^{neg} cells outside dermal condensate are likely undifferentiated migrating melanoblasts (i.e., MELA-A/B subpopulations) (Fig. 4j). This spatial *Edn3*, *Ednrb*, *Dct* co-expression pattern is highly consistent with the scRNA-seq data (Fig. 4i). Thus, our RNAscope result confirms the novel CellChat prediction of Edn3-Ednrb signaling from DC cells to melanocytes, implying the roles of DC cells in inducing early-stage directed migration of melanocytes into placodes ahead of epithelial Edn2 signaling. It also confirms novel, predicted autocrine Edn3-Ednrb signaling within dermal condensate.

We have added these new results into the Results section in the revised manuscript (Page 12).

Minor concerns:

R2-5. *Cell clustering is a pre-requisite for cell-cell interaction prediction, but clustering results can be different depending on which clustering methods are used and which parameters are chosen. How does such uncertainty affect the outcome of ligand-receptor interactions?*

Response: That is a good point. While different number of cell clusters may naturally affect the inferred ligand-receptor interactions, with a fixed cluster number the clustering results using different methods or parameters will unlikely have major impact on ligand-receptor interactions. This is because our cell-cell communication is inferred at the cluster level, only depending on estimation of the average gene expression in each cell cluster.

We demonstrated these two points using an example of E14.5 mouse embryonic skin dataset with four spatially colocalized cell subpopulations (i.e., placode, preDC, DC1 and DC2 in Figure 4). First, we assessed how cell clustering affects the inferred interactions if the number of cell clusters remains the same. We used two different choices of parameters (e.g., different number of highly variable genes and principle components) to produce two different clustering results while keeping the number of cell clusters unchanged. The Jaccard similarities between these two new clustering results and our original clustering result were 0.91 and 0.83, respectively. We then re-run CellChat analysis and found that all of the inferred interactions from our original clustering result were also predicted using these two newly added clustering results (Supplementary Fig. 16). Second, we used another choice of parameters to produce different number of cell clusters (three subpopulations: placode, preDC, DC). Applying CellChat to these three subpopulations, we found that 88% of interactions inferred from our original clustering result were also predicted using new clustering result (Supplementary Fig. 16). In general, the cell clustering needs to be carried out carefully in order to capture biologically meaningful cell populations before cell-cell communication analysis. We have added this in the revised Discussion (Page 24) and Supplementary Text (Page 6).

R2-6. *Related to the previous comment, nine fibroblast cell types were identified in mouse skin wound tissue. Are they truly distinct cell types? FIB-D seems to have unique signaling properties than the others. Have such specialized fibroblast cell types been observed before?*

Response: These nine fibroblast subclusters represent distinct fibroblast states, which were characterized by distinct marker genes (Supplementary Fig. 1d). In our original study (Guerrero-Juarez et al., *Nat Commun* 2019), we observed highly heterogeneity of wound fibroblasts using immunostaining. In particular, *Crabp1*-positive cells were validated to be enriched in upper wound dermis. FIB-D cells were enriched for high *Crabp1* expression and cell cycle genes, such as *Cenpa* (Supplementary Fig. 1d). Thus, it likely represents an actively cycling subset of *Crabp1*-positive cells. We have added this point in the revised Results section (Page 7).

R2-7. *The analysis of incoming and outgoing signaling patterns seems interesting, but it is unclear what is the distinction between incoming and outgoing patterns. Is there a mechanistic interpretation for these patterns?*

Response: Outgoing patterns reveal how the sender cells (i.e. cells as signal sources) coordinate with each other as well as how they coordinate with certain signaling pathways to drive communication. Incoming patterns show how the target cells (i.e. cells as signal receivers) coordinate with each other as well as how they coordinate with certain signaling pathways to respond to incoming signals. Such pattern analysis uncovers the coordinated responses among different cell types within the same tissue microenvironment. Different cell types may simultaneously activate same cell-type-independent signaling patterns or may also activate different cell-type-specific signaling patterns. This analysis can potentially help to derive general cell-cell communication principles. We added these details in the revised Results (Page 5) and Methods (Page 30).

R2-8. *False positives rate is used to evaluate the performance of italk and singlecellsignalR. How is this calculated exactly? The first sentence on page 27 is vague and seems to be associated with consistency rather than accuracy. What is the ground truth? What is the false positive rate of the method presented in this study? Why is CellPhoneDB not compared in this analysis?*

Response: We apologize for the confusion. Both CellChat and CellPhoneDB consider multi-subunit structure of ligands and receptors to represent heteromeric complexes accurately. This is critical, because cell-cell communication relies on multi-subunit protein complexes. The purpose of computing false positive rate here is to evaluate the effect of neglecting multi-subunit structure of ligands and receptors. Thus, we did not compute such false positive rate for both CellChat and CellPhoneDB, and we only evaluate the performance of tools that use only one ligand and one receptor gene pairs, such as SingleCellSignalR and iTALK. The false positive interactions are defined by the

interactions with multi-subunits that are *partially* identified by iTALK and SingleCellSignalR. The ground truth is based on our curated CellChatDB database. For example, for Tgfb1 ligand and its heteromeric receptor Tgfr1/Tgfr2 curated in CellChatDB, if the method only identifies one of the two pairs (Tgfb1-Tgfr1 and Tgfb1-Tgfr2), then we consider this prediction as one false positive interaction. Due to the lack of ground-truth for the inferred cell-cell communication network, we did not report the false positive rate of CellChat. However, CellChat only produces significant interactions on the basis of a statistical test that randomly permutes the group labels of cells. An empirical p-value is computed for each ligand-receptor pair for any two given cell groups. Therefore, CellChat likely has a good control of the false positive rate. In the revised manuscript we clarified these essential details (see Methods section; Page 35). We also clarified the point that our subsampling analysis was used to evaluate the consistency rather than accuracy.

R2-9. *On a practical side, the cellchat website posts predicted results in mouse skin, but a user might be interested to apply this method to analyze their own data. Is this possible?*

Response: Yes, it is possible. We envision the CellChat website will grow rapidly to become a community-driven web portal for a broad range of tissues as more datasets added to the website. For any given scRNA-seq dataset that has been processed by our R toolkit CellChat, we will host the predicted results on our server, allowing easy exploration and comparison of the cell-cell communication. We've added this point in the revised Discussion (Page 23).

Reviewer #3

In the manuscript Inference and analysis of cell-cell communication using CellChat, Jin et al. presented a database of interactions among ligand and receptors, and a methodology to infer inter-cellular communication networks. There are a few subsequent analyses based on the inferred network, including centrality related concepts like dominant senders/receivers/mediators, etc., as well as using non-negative matrix decomposition to break down the network into distinct patterns. The manuscript has provided the community an alternative way and great visualization tools for inferring and comparing cell-cell communication between different biological conditions using scRNA-seq data. However, as a methodology paper, some data provided in the manuscript should be further validated to support their conclusions and need to be carefully addressed.

Response: We thank the reviewer for the appreciation of the novelty of CellChat and the insightful comments to strengthen the method. Our detailed responses are provided below. Substantial improvement has been made in the revision, and multiple changes highlighted with red were introduced throughout the manuscript.

R3-1. *Lies at the core is an ambitious model law-of-mass-action model that takes into account almost everything: the ligand/receptor expression and what they form multi-units complexes, co-stimulatory, and co-inhibitory ligands/receptors, roles of agonist and antagonist. While all these components and their interactions are relevant, modeling of all these mechanistic processes requires a high level of details which is very hard to achieve with scRNA-seq data. Remember LMA happens in protein level, not in RNA level. So ironically, the model seems to capture everything but there are many assumptions, almost all parameters are arbitrarily chosen, and not easy to justify. For instance, why the dissociation constant is always 0.5? Why the Hill coefficient is always 1? Such a detailed model will make the study a lot more depends on the correctness of the curated database.*

Response: This is an important point. We agree that a high level of details (e.g., protein levels in individual cells) will improve the modeling of all these components. The

framework presented in this study will likely motivate further study once other types of omics data are available. Due to the technical difficulties of capturing single-cell proteomic information at present time, a comprehensive understanding of ligand-receptor interactions remains challenging. mRNA levels have been often used to estimate the level of proteins in many previous studies, and could be used to provide rough estimate of protein-protein interactions. In the revised manuscript, we added one paragraph in the Discussion to point out the main assumption of our modeling (Page 23).

We agree that as a data-driven approach, it is indeed difficult to determine a set of biologically meaningful parameters in the Hill function, in particular that different pairs of ligands and receptors often have different dissociation constants and different degree of cooperativity. Although it lacks direct or explicit biologically connections with the data, the Hill function used in our current model can be considered as a nonlinear approximation of the ligand-receptor interactions. We made this point clearer in the revised Discussion (Page 23).

To study how the choices of those parameters may affect the inferred ligand-receptor interactions, in the revision we varied the parameter values within certain ranges to explore the robustness of our method (new Supplementary Fig. 14). In particular, we varied the dissociation constant K_h from 0.1 to 0.9 with an increment of 0.2, and then computed the Jaccard similarity between the interactions inferred with each varied K_h and the interactions inferred with K_h being 0.5. We found the inference is relatively robust to the choice of K_h for all the four tested datasets. Similarly, by varying Hill coefficient n from 0.5 to 4, we also found the inferred ligand-receptor interactions are relatively robust. We added these robustness analyses in the revised Discussion (Page 23).

R3-2. *There's no doubt the authors did their best to construct CellChatDB, but there's no perfect source of information. I think, at the very minimal, the authors should do the following test to show that the outputs of their model capture a certain level of real signals rather than purely noise: In several ways randomize their curated database, like ligand-receptor interactions, the corresponding agonist/antagonist, co-receptors, etc., one-by-*

one and in some combinations, and then repeat the identification of statistically significant communications. If the number of significant pairs identified using the fake database is similar, then bad news, suggesting the outputs are simply false positive. In fact, the procedure could provide a way to quantify the false discovery rate.

Response: Thank you for the nice suggestion. In the revision we took the four spatially colocalized cell subpopulations in the E14.5 embryonic skin dataset (i.e., placode, preDC, DC1 and DC2 in Figure 4) as an example for this analysis. In particular, we computed the ratio of the number of inferred interactions from a randomized database over the number of inferred interactions from the original curated database. When randomizing ligands, receptors, the combination of ligands and receptors (denoted by ligands and receptors) and the combination of either ligands or receptors and one of the co-factors (such as agonist, antagonist and co-receptors) for 50 times, we found that the ratio values were 63% on average (see the inclined Figure 1a). When randomizing the agonist, antagonist, co_A_receptors and co_I_receptors independently, the number of inferred interactions was almost the same as the real curated database. This is not surprising because the agonist, antagonist, co_A_receptors and co_I_receptors are considered as co-factor molecules, which only modulate ligand-receptor mediated signaling strength either positively or negatively. Therefore, randomizing these co-factors does not have prominent influence on whether ligand-receptor pairs are significant or not. Moreover, the percent of ligand-receptor pairs with co-factors in CellChatDB is only about 20%. However, we observed a significant reduction in the inferred interaction strength when randomizing antagonist, co_A_receptors and co_I_receptors (see the inclined Figure 1b), suggesting the role of co-factors in modulating the interaction strength.

Moreover, we randomized the ligands, receptors, the combination of ligands and receptors in the database of other methods including CellPhoneDB, iTALK and SingleCellSignalR. The ratios of inferred interactions between randomized databases and real databases for these three methods were about 69%, 83% and 77% respectively, which is higher than the computed ratio value from CellChat (see the inclined Figure 1c). For these methods, the relatively higher number of inferred interactions in the randomized

database was possibly caused by the following four reasons: 1) Multiple ligand-receptor pairs usually contribute to the communication between two cell types; 2) Many ligands and receptors are not uniquely expressed in one cell type; 3) Many ligands (or receptors) may share the same receptors (or ligands), such as WNT signaling and cytokine/chemokine signaling; 4) A fake pair of ligand and receptor likely contributes to the communication for an emerging combination of two cell types. For example, for a fake pair of ligand LB that is highly enriched in cell type B1 and receptor RA that is highly enriched in cell type A1, a fake pair of ligand LB and receptor RA likely contributes to the communication between cell types B1 and A1. The fourth reason might be the main reason why randomizing database still produces a high number of significant ligand-receptor interactions. We did not add these analyses and comments in the revised manuscript, but we are happy to include it if the reviewer prefers such new addition.

Inclined Figure 1. The number of the inferred interactions using randomized databases. **(a)** The ratio of the number of inferred interactions from a randomized database over the number of inferred interactions from the original curated CellChatDB database. The inferred interactions are from CellChat. We randomized ligands, receptors, the combination of ligands and receptors (denoted by ligands & receptors), the combination of either ligands or receptors and one of the co-factors (such as agonist, antagonist and co-

receptors), agonist, antagonist, co_A_receptors and co_I_receptors for 50 times. Bar plots show the mean value and the standard error (indicated by the error bar). **(b)** Left panel: The percent of ligand-receptor interactions with co-factors in the CellChatDB database. Right panel: Comparison of the interaction strength (i.e., the communication probabilities) inferred by the curated CellChatDB with that inferred by the randomized databases. p-values are from two-sided Wilcoxon rank-sum tests. **(c)** Comparison of the number of inferred interactions among different methods using randomized databases. The ratios of the number of inferred interactions from a randomized database over the number of inferred interactions from the original database were presented. The ligands, receptors and the combination of ligands and receptors were randomized in the database provided by each method. Bar plots show the mean value and the standard error (indicated by the error bar). p-values are from two-sided Wilcoxon rank-sum tests.

R3-3. *In revealing continuous cell lineage-associated signaling events, the authors predict the cell-cell communication in the different stages during pseudotime and real embryonic stages (E13.5/E14.5). However, the authors should provide expression pattern of ligands and receptors in all the predicted interactions during skin cell development side by side to validate their prediction results. Based on methods, we should be able to see a similar pattern of expression of ligand/receptors with communication probability of predicted interactions during developmental stages.*

Response: Thank you for the nice suggestion. In the revision we have created a stacked violin plot to show the expression patterns of ligands and receptors. We presented the expression patterns of related signaling in both main and supplementary figures, and found similar patterns of expression of ligands/receptors with the communication probability (updated Figures 3-5 and Supplementary Figs. 3 and 5). In addition, the average expressions of all predicted ligands and receptors in each cell subpopulation are shown in the newly added Supplementary Data 2.

R3-4. *In predicting key signaling events between spatially colocalized cell populations, the authors used spatially-colocalized 4 cell types to showcase their prediction on the cell-cell communication. However, proper controls are not provided to validate the predictions. The authors should add in cell types that are not spatially adjacent to these 4 cell types to the same analysis and demonstrate that cell-cell communication identified in Figure 4 are stronger in spatially adjacent cell types but not in spatially distant cell types.*

Response: Thank you for the good suggestion. In the revision we now added seven cell types from the E14.5 embryonic skin dataset: FIB (fibroblasts), MELA (melanocytes), Spinous (spinous epithelial cells), MYL (myeloid cells), Immune (other immune cells), ENDO (endothelial cells) and Muscle. We have computed the number of inferred communications as well as the sum of communication probabilities between each cell type and for the four spatially colocalized cell populations. We found that CellChat consistently captured stronger interaction in spatially adjacent cells than in spatially distant cells both in terms of the number of interactions and the interaction probabilities (new Supplementary Fig. 9a-b). In addition, by taking into an account all seven cell types together, we found that CellChat can significantly distinguish the spatially adjacent from spatially distant cells (new Supplementary Fig. 9c). We have added these new results into the Results section in the revised manuscript (Page 20).

In addition, using multiplexed RNA *in situ* detection (RNAscope), we have now experimentally validated two novel CellChat predictions (that were not previously described in the literature) on the signaling interactions during early hair follicle morphogenesis in developing embryonic skin. First prediction: CellChat revealed that at E14.5, DC cells respond to autocrine PROS pathway (Fig. 4g). Pros1 is the ligand for the pathway, that signals via receptor tyrosine kinase Axl. Signaling via Axl has been implicated in conferring cells with migratory properties in different biological context, including cancer invasion^{44, 45} and directional migration has been recently shown to be crucial for normal dermal condensate formation upon hair follicle morphogenesis⁴⁰. We examined CellChat's prediction of active PROS signaling in DC cells by RNAscope for *Edn3* as DC marker, *Axl* and *Thy1* (Cd90) as the marker of cell migration^{46, 47}. As expected from scRNA-seq, *Axl* was expressed broadly, including in *Edn3*+ DC, overlaying placode and surrounding epithelium. However, *Thy1* expression was concentrated in DC with significantly lower levels elsewhere (Fig. 4h). This RNAscope result is consistent with the possibility of autocrine PROS signaling in DC, likely driven via Pros1-Axl signaling.

Second prediction: During early hair follicle formation at E14.5, melanoblasts (melanocyte precursor cells) migrate into the hair placode from the dermis and then

becomes differentiated toward melanocytes. However, the mechanisms of melanocyte migration into placode remain incompletely understood⁴⁸. Therefore, we further studied the cell-cell communication among placodes, DC cells and melanocyte cells (including three melanocyte subpopulations: MELA-A, -B and -C; see Methods). CellChat revealed that melanocytes strongly respond to DC cells via previously unrecognized EDN signaling (Fig. 4i). *Edn3* is the ligand for EDN pathway, that regulates melanocytes migration⁴⁹. Therefore, CellChat prediction suggests DC cells induce early directed migration of melanocytes. To experimentally examine this prediction, we used RNAscope technique to spatially map expression of *Dct*, that marks late-stage melanocyte precursors, *Edn3* ligand and its receptor *Ednrb* in E14.5 embryonic mouse skin. As expected, *Dct*⁺ melanocytes (i.e., MELA-C subpopulation) localize inside the placode. They also express *Ednrb*. In turn, *Edn3* is specifically enriched in DC cells (preDC, DC1 and DC2 subpopulations), while *Ednrb* is also enriched in a portion of DC cells (likely DC2 subpopulation). Scattered *Ednrb*⁺/*Edn3*^{neg}/*Dct*^{neg} cells outside dermal condensate are likely undifferentiated migrating melanoblasts (i.e., MELA-A/B subpopulations) (Fig. 4j). This spatial *Edn3*, *Ednrb*, *Dct* co-expression pattern is highly consistent with the scRNA-seq data (Fig. 4i). Thus, our RNAscope result confirms the novel CellChat prediction of *Edn3*-*Ednrb* signaling from DC cells to melanocytes, implying the roles of DC cells in inducing early-stage directed migration of melanocytes into placodes ahead of epithelial *Edn2* signaling. It also confirms novel, predicted autocrine *Edn3*-*Ednrb* signaling within dermal condensate. We have added these new results into the Results section in the revised manuscript (Page 12).

R3-5. *In comparison with other cell-cell communication inference tools, current metrics used to compare the tools by reasoning that a more accurate method will have a larger proportion of overlapped predictions with other methods is not reasonable and the result is not convincing. The ligand receptor databases used in CellChat and CellPhoneDB are different which will directly contribute to the number and variety of predicted interactions. Besides, the cell interactions identified by SingleCellSignalR and iTalk but not CellChat, due to failed detection of interactions with multi-subunits, are not necessarily 'false-*

positive', which could be caused by low expression of multi-subunits of the receptors. The authors should use better metrics to compare those inference tools, for example, whether these tools can correctly capture stronger interaction in spatially adjacent cells but not spatially distant cells. In Supplementary Figure 8(a), the authors overlapped ligand-receptor interactions between CellChat/CellPhoneDB and other two methods including SingleCellSignalR and iTALK. CellChat should be compared with CellPhoneDB in terms of overlapping interaction. In Supplementary Figure 8(b), CellChat is not outperforming CellPhoneDB much even with the modeling of almost everything. The authors need to explain this in "Method comparisons" section.

Response: Thank you for the insightful comments and suggestions. We agree with the reviewer's point on the evaluation metrics used in our original manuscript. We have clarified these points in the revised Results (Page 18) and Discussion (Page 25).

In the above response to R3-4, we mentioned that we have now added seven cell types that are likely spatially distant from Placode and DC cells within E14.5 embryonic skin dataset. Here we tested whether these tools can correctly capture stronger interaction in spatially adjacent cells (i.e., Placode and DC cells). Generally, CellChat is able to capture stronger interaction in spatially adjacent vs. spatially distant cells both in terms of the number of interactions and the interaction probabilities (new Supplementary Fig. 9a-b). We found that CellPhoneDB also performed well in discriminating the spatially adjacent from spatially distant cells. On the other hand, iTALK failed to capture stronger interactions in the spatially adjacent cells for FIB, MELA, MYL and ENDO. SingleCellSignalR failed to capture stronger interactions in spatially adjacent cells for FIB and ENDO. In addition, by taking into an account all these seven cell types together, we found that both CellChat and CellPhoneDB can significantly distinguish the spatially adjacent cells from spatially distant cells, whereas iTALK and SingleCellSignalR failed to do so (new Supplementary Fig. 9c). Since CellPhoneDB infers more interactions than CellChat, we tested whether the top interactions predicted by CellPhoneDB can also distinguish the spatially adjacent cells from spatially distant cells. For the top 10%, top 20% and top 30% interactions predicted by CellPhoneDB, the differences between

spatially adjacent and spatially distant cells were not as significant as seen when using CellChat (new Supplementary Fig. 10a-b), suggesting that CellChat performed better at capturing stronger interactions. Together, although CellChat produces fewer interactions, it performs well at predicting stronger interactions. These results have been added into the revised Results (Page 20).

In the revision we have also computed the overlapping interactions between CellChat and CellPhoneDB for any two cell groups and found that these two methods share about 50% interactions (updated Supplementary Figure 8a). The results of Supplementary Figure 8b assessed the consistency of the inferred interactions when subsampling the cells from the data. Both CellChat and CellPhoneDB are relatively robust to subsampling, which is likely because both methods infer cell-cell communication based on cell clusters. We have added this point in the revised manuscript (Page 19).

R3-6. *The authors adopted non-negative matrix factorization for the identification of major signals of specific cell groups and global communication patterns. The number of patterns 5 seems to be random or experiential. Without knowing pattern's biological meaning, it is unrealistic to guess the real number of patterns even with the domain knowledge. In page 10, the authors claimed that they can predict the sequential signaling events of cells, e.g., FIB-A cell secreted EGF and GALECTIN signals first. Then FIB-D and FIB-E coordinate... It is easier to identify groups of cell types and signals, but how this time-series event is inferred from the patterns is unclear.*

Response: Thank you for pointing this out. In the revision we now have inferred the number of patterns based on two metrics that have been widely used in the literature and implemented in the NMF R package, including Cophenetic and Silhouette. Both metrics measure the stability for a particular number of patterns based on a hierarchical clustering of the consensus matrix. For a range of the number of patterns, a suitable number of patterns is the one at which Cophenetic and Silhouette values begin to suddenly drop. By applying these two metrics to the three dataset we studied, we found that the inferred number of patterns was ranging from 4 to 6 for the wound dataset and E14.5 pseudotime

dataset (new Supplementary Fig. 15a-b). For the outgoing communication patterns in the wound dataset and the incoming communication patterns in the E14.5 pseudotime dataset, these two metrics predicted that the number of patterns was 4 and 6, respectively. Comparing to the five outgoing communication patterns in the wound dataset (Fig. 2g), the four outgoing communication patterns merged two fibroblast-related patterns into one pattern. Specifically, FIB-H and other three fibroblast subpopulations (FIB-D, FIB-F and FIB-I) that were originally associated with two different patterns (Fig. 2g) were now associated with a single pattern (Supplementary Fig. 15d). For the E14.5 embryonic DC_Placode dataset with four spatially colocalized cell populations, the two metrics predicted that the number of incoming communication patterns was two (Supplementary Fig. 15c). Compared to the three incoming communication patterns (Fig. 4f), the two incoming patterns merged the pre-DC-enriched pattern with the DC-enriched pattern (Supplementary Fig. 15e). In another words, pre-DC and DC that were originally enriched in two different patterns (Fig. 4f) were now enriched in a single pattern (Supplementary Fig. 15e). Therefore, these different numbers of patterns provided a different resolution to uncovering the coordinated responses among different cell types. CellChat R package now provides a visual representation of Cophenetic and Silhouette metrics for a range of the numbers of patterns to enable users select the optimal setting for the number of patterns present in the dataset. We added these results in the revised Discussion (Page 23) and Supplementary Text (Page 5).

To predict the sequential signaling events, we combined the communication pattern analysis with the inferred pseudotemporal cell events. The dermal and epidermal trajectory analysis potentially revealed the pseudotemporal order of different cell types, and the communication pattern analysis identified strong signals that were sent or received by certain cell types. Therefore, the combination of these two analyses allows to potentially uncover sequential signaling events. We have made this point clearer in both the revised Results (Page 10).

Minor points:

R3-7. While P_{ij}^k is likely to lie between 0 to 1 (because of the last term), it is not exactly a probability, in the sense, $\sum_j P_{ij}^k$ might not be 1. Should there require certain normalization? Or simply say the quantity scales with the probability?

Response: Thanks for raising this point. We did not perform normalization along the second dimension because the normalized data are not suitable for comparing the interaction strength between different cell types across multiple signaling pathways. In the revised manuscript, we emphasize that this quantity measures the communication strength and we use it to represent the communication probability in revised Methods (Page 28).

R3-8. Because of dropout, 0 is quite common in scRNA-seq data. I am slightly worried about estimating the level of ligands by the geometric mean of the sub-units. Similarly, for the robust measure of average gene expression based on Q1, Q2, Q4. So, if there are more than 25% of dropout, EM=0. Can the authors provide some statistics on how many pairs are dropped?

Response: We agree that dropout events often occur in scRNA-seq data due to the low amounts of mRNA in individual cells. Previous study showed that dropouts likely happen for genes with lower expression magnitude instead of high expression magnitude (Kharchenko et al., Nat Methods, 2014). Therefore, dropouts will not likely affect the strong signals predicted by CellChat.

In the revision we have now systematically explored the inferred ligand-receptor pairs using different methods by calculating the average gene expression per cell group, including mean (i.e., simply calculating the average gene expression), 5% truncated mean (i.e., calculating the average gene expression by discarding 5% from each end of the data), 10% truncated mean, trimean (i.e., the method used in CellChat) and median. For the four studied datasets, there are about 15% more dropped ligand-receptor pairs when calculating the average gene expression using trimean compared to the 10%

truncated mean (Supplementary Fig. 17). Compared to other cell-cell communication tools, such as CellPhoneDB that uses 10% truncated mean, CellChat produces fewer ligand-receptor interactions. As seen in our added study on the spatially adjacent subpopulations (Supplementary Fig. 10a-b), CellChat performs well at predicting stronger interactions, which is helpful for narrowing down on interactions for further experimental validations. In the CellChat R package, users can now calculate the average gene expression per cell group using these different methods. We discuss these two points in the revised Discussion (Page 24).

R3-9. About the non-negative matrix factorization step, the authors reduce the 3D array P to 2D by summing over the receivers so that NMF could be used. It erases patterns associated with the receiver-end. What happens if we sum over the sources? Have the authors considered tensor decomposition?

Response: In our original manuscript, we performed two communication pattern analyses, including for outgoing and incoming communication patterns. Outgoing patterns were found for the sender cells (i.e. cells as signal sources) by summarizing the communication probability array P (three-dimensional) along the second dimension, and the incoming patterns were found for the target cells (i.e. cells as signal receivers) by summarizing the communication probability array P (three-dimensional) along the first dimension. We have made this point clearer both in the revised Results (Page 6) and Methods (Page 30).

When uncovering the coordinated responses among different cell types, we did not use the tensor decomposition partly because it is not applicable to the following two situations: 1) the number of outgoing and incoming patterns could be different and 2) The coordinated signaling pathways could be different for secreting and receiving cells. In other words, secreting cells could send a group of certain signaling pathways and receiving cells could respond to another group of certain signaling pathways. To demonstrate these two points, we took the four spatially colocalized cell subpopulations in the E14.5 embryonic skin dataset (i.e., placode, preDC, DC1 and DC2 in Figure 4) as an example. For the first point, as shown in the response to the comment R3-6, the

Cophenetic and Silhouette metrics predicted that the number of outgoing and incoming communication patterns was 3 and 2 respectively (Fig. 4e and Supplementary Fig. XX). To demonstrate the second point, we applied the Non-negative Tucker Tensor Decomposition (NTD) algorithm to the three-dimension communication probability array P using nnTensor R package. First, we performed NTD with rank 2, producing 2 outgoing and 2 incoming patterns (see the inclined Figure 2a). Obviously, the signaling pathways associated with each outgoing pattern are the same as the corresponding incoming pattern, which is usually not true. For example, the predicted patterns showed that preDC/DC1/DC2 cells send signal PROS and placode cells respond to signal PROS, which is not consistent with the signaling network predicted by CellChat (see the inclined Figure 2c). We observed the same issue when performing NTD with rank 3, which incorrectly predicted preDC/DC1/DC2 cells as major sources of TGFb signaling (see the inclined Figure 2b and 2c). Therefore, compared to the tensor decomposition approach, the matrix decomposition approach we used provides a more biological meaningful way to identify the outgoing and incoming communication patterns. We have now mentioned this point in the Discussion section (Page 24) and Supplementary Text (Page 5).

Inclined Figure 2. The outgoing and incoming patterns predicted by the Non-negative Tucker Tensor Decomposition (NTD). **(a)** NTD algorithm was applied with rank 2, producing two outgoing and incoming patterns. This alluvial plot shows the correspondence between the inferred latent patterns and cell groups as well as signaling pathways. The thickness of the flow indicates the contribution of the cell group or signaling pathway to each latent pattern. The height of each pattern is proportional to the number of its associated cell groups or signaling pathways. Outgoing patterns reveal how the sender cells coordinate with each other as well as how they coordinate with certain signaling pathways to drive communication. Incoming patterns show how the target cells coordinate with each other as well as how they coordinate with certain signaling pathways to respond to incoming signaling. **(b)** NTD algorithm was applied with rank 3, producing three outgoing and incoming patterns. **(c)** Example signaling networks predicted by CellChat.

Circle sizes are proportional to the number of cells in each cell group and edge width represents the communication probability.

R3-10. The standard of good figure legends is that the readers can easily get an idea of the figures without going back and forth among main text, figures and methods. The authors should improve their figure legends and clearly demonstrate what they did in each figure, instead of just generally saying what kind of plot/diagram it is.

Response: This is a good suggestion. We have now comprehensively revised our figure legends in the revision.

REVIEWERS' COMMENTS<

Reviewer #1 (Remarks to the Author):

The authors have improved the manuscript and CellChat in multiple ways. I have only one additional minor suggestion below.

Given the wealth of references rapidly emerging in this domain, it is hard to blame the authors for providing an exhaustive bibliography: however, CellTalkDB (PMID: 33147626) also contains mouse ligand-receptor pairs, which should be compared. Moreover, inferring cell-cell communication through single-cell transcriptomics data has been systematically reviewed (PMID: 33168968, 32435978), which should be mentioned in Introduction as well.

Reviewer #2 (Remarks to the Author):

The authors have done a great job in revising their manuscript and addressed all the issues I raised satisfactorily.

Reviewer #3 (Remarks to the Author):

My questions were well addressed. This tool will be very useful for the community.

REVIEWERS' COMMENTS

Reviewer #1

The authors have improved the manuscript and CellChat in multiple ways. I have only one additional minor suggestion below.

Given the wealth of references rapidly emerging in this domain, it is hard to blame the authors for providing an exhaustive bibliography: however, CellTalkDB (PMID: 33147626) also contains mouse ligand-receptor pairs, which should be compared. Moreover, inferring cell-cell communication through single-cell transcriptomics data has been systematically reviewed (PMID: 33168968, 32435978), which should be mentioned in Introduction as well.

Response: We thank the reviewer for the insightful comments and for pointing out these important papers. In this revision, we added the comparison between CellChatDB and CellTalkDB (updated Supplementary Figure 1b) in both human and mouse. For the database in human, the number of L-R pairs in CellChatDB (i.e., 1,939 pairs) is lower than CellTalkDB (i.e., 3,398 pairs); for the database in mouse, the number of L-R pairs in CellChatDB (i.e., 2,021 pairs) is comparable with CellTalkDB (i.e., 2,033 pairs). However, CellTalkDB does not consider L-R pairs with multi-subunits as well as the cofactors. Taking into an account the subunit structure of ligands and receptors is essential because cell-cell communication often relies on multi-subunit protein complexes. Cofactors also modulate cell-cell communication both positively and negatively for certain signaling pathways. Therefore, CellChatDB provides an important resource for identifying biologically meaningful cell-cell communication. We added these results in the revised manuscript and updated Supplementary Figure 1b. Moreover, we cited the two references (PMID: 32435978, 33168968) in the Introduction, which correspond to Ref. [6] and [7] in the revised manuscript.

Reviewer #2

The authors have done a great job in revising their manuscript and addressed all the issues I raised satisfactorily.

Response: We thank the reviewer for the insightful comments and suggestions that helped us improve the manuscript.

Reviewer #3

My questions were well addressed. This tool will be very useful for the community.

Response: We thank the reviewer for appreciating the importance of our method and for the insightful comments.